# TPE: Towards Better Compositional Reasoning over Conceptual Tools with Multi-persona Collaboration

## Abstract

Large language models (LLMs) have demonstrated exceptional performance in planning the use of various *functional tools*, such as calculators and retrievers, particularly in question-answering tasks. In this paper, we expand the definition of these tools, centering on *conceptual tools* within the context of dialogue systems. A *conceptual tool* specifies a cognitive concept that aids systematic or investigative thought. These *conceptual tools* play important roles in practice, such as multiple psychological or tutoring strategies being dynamically applied in a single turn to compose helpful responses. To further enhance the reasoning and planning capability of LLMs with these *conceptual tools*, we introduce a multi-persona collaboration framework: Think-Plan-Execute (**TPE**). This framework decouples the response generation process into three distinct roles: Thinker, Planner, and Executor. Specifically, the *Thinker* analyzes the internal status exhibited in the dialogue context, such as user emotions and preferences, to formulate a global guideline. The *Planner* then generates executable plans to call different *conceptual tools* (e.g., sources or strategies), while the *Executor* compiles all intermediate results into a coherent response. This structured approach not only enhances the explainability and controllability of responses but also reduces token redundancy. We demonstrate the effectiveness of **TPE** across various dialogue response generation tasks, including multi-source (FoCus) and multi-strategy interactions (CIMA and PsyQA). This reveals its potential to handle real-world dialogue interactions that require more complicated tool learning beyond just *functional tools*. The full code and data will be released for reproduction.

## 1 Introduction

Tools have been briefly defined as an object carried or maintained for future use[1] (Finn et al., 2009), which serve as indispensable extensions of human capabilities, enhancing productivity, efficiency, and problem-solving (Washburn, 1960). Their creation and utilization stem from the fundamental desire to overcome physical limitations and pioneer new frontiers, especially in the era of the digital world (Qin et al., 2023a). Since the advent of Large Language Models (LLMs), such as ChatGPT, tool-oriented AI research has become hot by exploring their planning capability to call different tools to solve complex questions (Hsieh et al., 2023; Lu et al., 2023; Xu et al., 2023a). However, most of these tools fall into one specific category: *functional tool* (Jones & Kamil, 1973), such as APIs (Li et al., 2023b; Qin et al., 2023b), Models (Shen et al., 2023), Programs (Liang et al., 2023), and so on, which primarily are designed and used to perform specific tasks or functions effectively.

In this paper, we first expand the applicability of tools for LLMs by introducing *conceptual tool*, serving as another important type of tool in practice (Dye, 2011). A *conceptual tool* specifies a cognitive concept used to help systematic or investigative thought[2]. These *conceptual tools* are not functional but rather mental constructs or theoretical models used in many fields like philosophy (Sun et al., 2021), science (Brette et al., 2007), education Stasaski et al. (2020), and business (Hakala

---

[1]https://en.wikipedia.org/wiki/Tool
[2]We use a similar definition here following the description in the corresponding Wikipedia page: *By extension, concepts which support systematic or investigative thought are often referred to as "tools".*

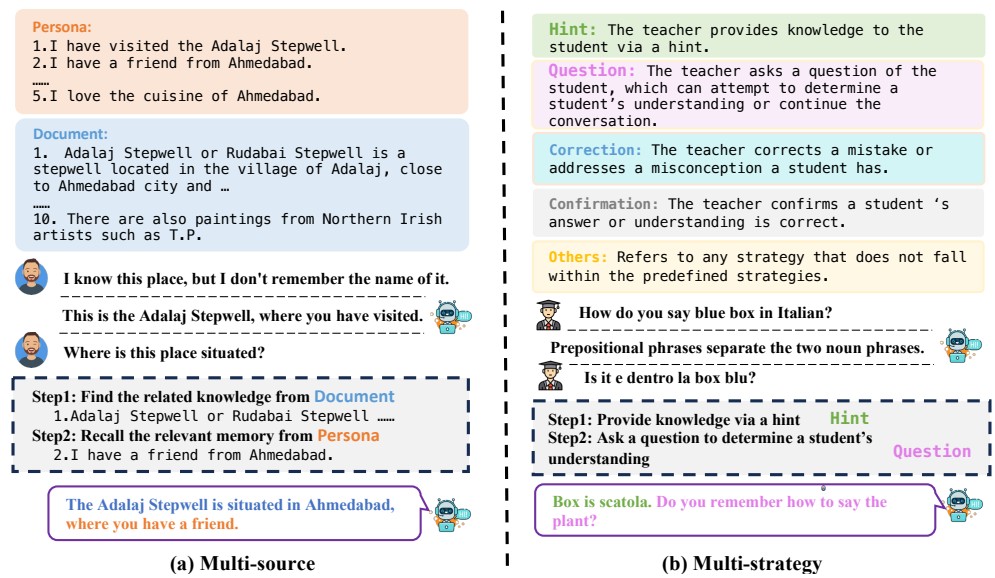

Figure 1: Two typical dialogue systems necessitating the planning capability of LLMs to 1) call different sources of knowledge; 2) call different strategies; in order to compose the final response being more personalized and helpful. The examples here are chosen from public datasets: FoCus (Jang et al., 2022) and CIMA (Stasaski et al., 2020) respectively. We employ color coding to denote various sources and strategies. To ensure clarity, we consistently use the same color in responses to signify their association with the respective sources or strategies.

& Vuorinen, 2020) to facilitate thinking, problem-solving, and communication. For example, a decision-making process "developed to help women and their partners make confident and informed decisions when planning where to give birth" is described as a *conceptual tool* for birth choice. Similarly, reflection serves as a *conceptual tool* to enhance the professional development of trainee teachers (Dye, 2011).

Such *conceptual tools* are important and common, especially in the context of dialogue systems. Considering two typical conceptual tools: source and strategy during dialogue response generation (shown in Figure 1), given the dialogue context in multi-source dialogue, the system needs to take two steps: 1) first retrieve related knowledge from the *Document* source to get the location; and then 2) retrieve related personal memory or experience from *Persona* source, to provide personalized and informative responses. The key concern here is whether or not LLMs can comprehend the distinctions between the concepts of *Persona and Document* sources and determine the correct order to call them[3], rather than focusing on the *functional tool*, such as a retriever. Besides different sources, individuals typically employ a range of strategies, whether singly or in combination, to furnish constructive and refined responses in practical contexts. These strategies are mostly cognitive in their utilization, particularly within domains such as negotiation (He et al., 2018; Zhou et al., 2019), debate (Wei et al., 2016; Wiegmann et al., 2022), psychological therapy (Sun et al., 2021; Cheng et al., 2022; Li et al., 2023a), and tutoring (Stasaski et al., 2020; Wang et al., 2023b; Macina et al., 2023a). For example, tutoring dialogue systems need to seamlessly blend educational content with motivational strategies to optimize the learning experience as shown in the right part of Figure 1. Similarly in psychological therapy, responses may require not only factual information but also empathetic communication with professional therapeutic strategies (Hill, 2009; Zheng et al., 2023). However, most previous works downplay the sequential and dynamic relationship of different sources (or strategies) (Stasaski et al., 2020; Jang et al., 2022), and some of them formulate these tasks as classification tasks or seq2seq tasks that necessitate further in-domain finetuning (Wang et al., 2023b; Zheng et al., 2023), which brings challenges for current LLMs. In this way, our research seeks to expand the scope of LLMs' toolsets, enabling them to dynamically and automatically compose responses that draw from a multitude of *conceptual tools*.

---

[3]The call order varies across different contexts, such as *Persona* first and then *Document* (Appendix A.2).

Inspired by recent progress that unleashes the cognitive synergist in LLMs with multi-persona collaboration (Wang et al., 2023c; Chen et al., 2023), we introduce a multi-persona framework: Think-Plan-Execute (a.k.a, **TPE**), a novel prompting paradigm to enhance the planning ability of *conceptual tools* for the dialogue system. Specifically, **TPE** involves a structured decomposition of the overall planning process into three distinct phases, managed by three separate roles: *Thinker*, *Planner*, and *Executor*. The *Thinker* reasons the *internal status* exhibited in the dialogue context considering the comprehensive linguistic cues underneath the multiple dialogue interactions (Mairesse et al., 2007; Wang et al., 2023a), such as the user's emotional states or preferences, and formulates a blueprint of plans, serving as a global guideline for the *Planner* and *Executor*. The *Planner* needs to generate specific and executable plans in a natural-language format to call different *conceptual tools* (sources or strategies), while the content varies across tasks. The last *Executor* strictly follows the thought of the *Thinker* and the plan of the *Planner* to execute, and assemble all intermediate results to compose the final response. Thanks to the collaboration and decoupling of different roles, **TPE** offers an improved response generation process characterized by enhanced explainability and controllability while mitigating the issue of redundant interleaved prompts in observation-dependent planning (Yao et al., 2023). Overall, our main contributions can be summarized as follows:

- We pioneer the introduction of *conceptual tools*, serving as a significant extension and a new perspective to address complex dialogue situations, particularly for multi-source and multi-strategy dialogues.
- We introduce **TPE**, a tailored and explainable top-down dialogue planning framework that guides the LLMs by considering the internal status during the dialogue context (*Think*), sequentially and dynamically planning sources or strategies (*Plan*), and generating final responses (*Execute*), thereby producing more explainable and personalized responses.
- We conduct extensive experiments and provide in-depth analysis on three datasets, spanning multi-source, tutoring strategies, and psychological strategies planning, to demonstrate the effectiveness and efficiency of **TPE**.

## 2 RELATED WORK

**Tool-augmented LLMs.** Taking advantage of in-context learning (Brown et al., 2020) and chain-of-thought (CoT) reasoning (Wei et al., 2022), LLMs have shown their effectiveness in planning the use of various *functional tools* to interact with the physical world, such as a retriever to sync the most up-to-date web/knowledge resources, a calculator to calculate the math problem (Qin et al., 2023a; Zhuang et al., 2023; Lu et al., 2023; Shen et al., 2023). Specifically, WebGPT (Nakano et al., 2021) and ReAct (Yao et al., 2023) use a single retriever to retrieve different types of knowledge from the internet, ignoring the source of knowledge. Furthermore, Chameleon (Lu et al., 2023) and ReWOO Xu et al. (2023a) extend the toolset to contain models (Shen et al., 2023), calculators, the Python program (Liang et al., 2023), and so on, in order to solve a complex question. However, all of these tools fall into one category, which focuses on the primary purpose or function, namely *functional tools*. We alternatively target the important but under-explored side: *conceptual tools*, which centers on the cognitive concepts to guide the reasoning and planning of LLMs (Xu et al., 2023b).

**Multi-source/strategy Dialogue System.** The dialogue system relies on a variety of sources and strategies to enhance the quality of responses, ensuring they are engaging (Yang et al., 2022), trustworthy (Bang et al., 2023), personalized (Jang et al., 2022; Fu et al., 2022), empathetic (Zheng et al., 2023), and encompassing other vital features, depending on the particular source or strategy employed. However, most of the previous methods for multi-source/strategy dialogue either simply consider single-source/strategy situations in one turn or ignore the complex relationship between multi-source/strategy. Specifically, some works apply retrieval over a single source knowledge base — Wikipedia to aid knowledge-intensive dialogue (Dinan et al., 2019; Komeili et al., 2022; Bao et al., 2022), and there are other works indiscriminatingly utilize all available sources of knowledge for each turn (Jang et al., 2022; Wu et al., 2021; 2022). Similarly, some work targeting tutoring dialogue systems assume that the strategy is provided to guide the response (Macina et al., 2023a;b), and most methods formulate the multi-strategy response generation as a seq2seq task Sun et al. (2021); Wang et al. (2023b). Setting ourselves apart from prior endeavors, we leverage the exceptional planning capability of LLMs to call *conceptual tools*, empowering the dialogue system to sequentially and flexibly invoke various sources or strategies, making it well-suited to handling complex transitions Zheng et al. (2023) in real-world scenarios.

## 3 METHOD

In this section, we first formulate a response generation task based on LLM incorporating our newly defined *conceptual tools*. These tools are specifically designed for interaction scenarios that require complex reasoning and planning, particularly in multi-source knowledge grounding and multi-strategy dialogues. Subsequently, we introduce a multi-persona framework Think-Plan-Execute (`TPE`) to augment the LLM's capabilities in reasoning and planning the complicated flow of dialogue response generation, including internal status reasoning (*Thinker*), source/strategy planning (*planner*), and response generation (*Executor*).

### 3.1 TASK DEFINITION

Given a dialogue context $\mathcal{C} = \{u_1, s_1, u_2, s_2, \ldots, u_i\}$, where $u_j$ and $s_j$ represent the user's and system's utterances at turn $j$, respectively, the objective of the dialogue system is to generate a final response, $s_i$, in response to the user query while maintaining consistency with the dialogue context. In contrast to previous approaches that focus on functional tools (denoted as $t^{func}$), we adopt an extended and specialized tool definition to meet the complex situations in practice by proposing that multiple cognitive concepts of external knowledge/world can be regarded as *conceptual tools* ($t^{concept}$). Thus we define a toolset, denoted as $\mathcal{T} = \{t_1 : \text{desc}_1^{tool}, t_2 : \text{desc}_2^{tool}, \ldots, t_n : \text{desc}_n^{tool}\}$, where $desc_j^{tool}$ denotes a description of a *functional/conceptual tool $t_j$*. These tools can be sequentially and dynamically employed to compose the final response (Lu et al., 2023; Xu et al., 2023a), $s_i$, being personalized (Jang et al., 2022), empathic Zheng et al. (2023), thought-provoking Cheng et al. (2022); Macina et al. (2023a) and so on depending on which tool is invoked.

### 3.2 THINK-PLAN-EXECUTE FRAMEWORK

To meet the needs of a dialogue system that tracks user status and coordinates multiple-source knowledge and strategies essential for effective system response generation, we propose a novel multi-persona framework called Think-Plan-Execute (`TPE`), which consists of three consecutive roles: *Thinker*, *Planner*, and *Executor* as illustrated in Figure 2. We describe the roles as follows.

***Thinker*** module analyzes the ongoing dialogue and user queries to deduce the current user's internal status, encompassing factors such as user preferences, emotional state, and needs, then further anticipates a blueprint to guide subsequent steps. We collectively denote the user's status and the associated blueprint as the `thought`. *Thinker* operates in a manner akin to a human, engaging in immediate situational analysis and generating initial ideas. Consider the left case illustrated in Figure 1 as an example. Following multiple interactions with the user, the user's status might be updated as *The user does not clearly remember the location of "Adalaj Stepwell", a place he has previously visited, and is now inquiring about its whereabouts.*" and the corresponding blueprint could be "*I need to search external documents to answer the question and access the user's personal memory to help him recall this information..*"

Formally, a `thought` consists of internal status, and a blueprint of the plan is generated as follows:

$$\text{thought} \leftarrow T(\mathcal{C}; \mathcal{D}_t; Per_t). \tag{1}$$

The $\mathcal{C}, \mathcal{D}_t, Per_t$ indicate dialogue context, demonstration, and persona of *Thinker* respectively[4]. The generated `thought` serves as a global guideline for directing the entire response generation process. Additionally, it can be employed to enrich the semantic information, complementing the original context. The *thought* serves as part of the query and intermediate reasoning results, facilitating the reasoning and planning of LLMs, in accordance with Yu et al. (2023); Wang et al. (2023a)

***Planner*** determines the specific sequence of sources or strategy calls based on the `thought` generated by *Thinker* and original context $\mathcal{C}$. Specifically, there are two distinct planning formats, corresponding to multi-source and multi-strategy respectively:

1) Consecutive tuples $(desc^{plan}, so(Q))$ where $desc^{plan}$ describes the current step, and $so(Q)$ represents a specific source to be searched for input $Q$. In this context, the LLM must not only determine the order of different sources but also consider the query dependency between sources. For example, it may use the retrieved personal memory as a query to retrieve related documents, or vice versa.

---

[4]We provide all prompt details in the Appendix B.6 to assure the reproductivity.

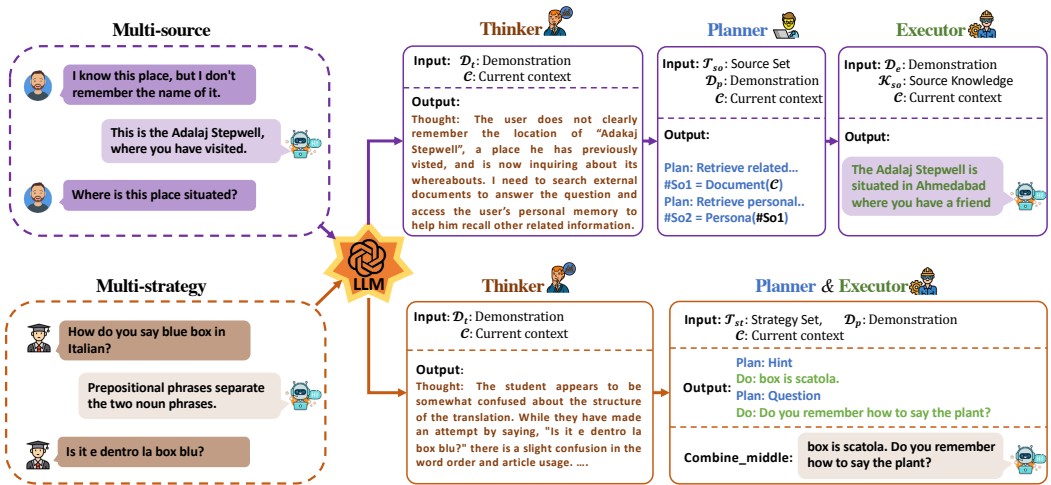

Figure 2: The workflow of our proposed think-plan-execute (**TPE**) framework, in which Thinker, Planner, and Executor are initialized using different personas with the same LLM. It's important to note that the Planner and Executor roles are undertaken by the same persona, who follows the guidelines by the Thinker under the multi-strategy scenario.

2) Consecutive $(st)$, where $st$ denotes a specific strategy name. For instance, as illustrated in the multi-strategy example in Figure 2, the *Planner* outputs $(Hint, Question)$ consecutively.

The plan is formulated as follows:

$$\text{plan} \leftarrow P(\mathcal{C}; \mathcal{D}_p, \mathcal{T}, Per_p), \tag{2}$$

where $\mathcal{D}_p, Per_P$ represent the demonstration and the persona of *planner*, respectively. Besides, $\mathcal{T}$ serves as documentation for multiple sources or strategies, providing a name and a description of each *conceptual tool*[5].

***Executor*** strictly follows the plan generated by *Planner*, calls *conceptual tools* (sources or strategies) sequentially, and composes intermediate results $\mathcal{M}$ into final responses. Consistently, there are two types of $\mathcal{M}$ in multi-source and multi-strategy respectively.

1) $\mathcal{M} = \mathcal{K} \leftarrow t_{func}(\mathcal{C}, so(Q))$, once the *Planner* decide the source to call at the current step, the functional tool – retriever is required to retrieve from corresponding sources. This retrieved knowledge from different sources serves as the intermediate results $\mathcal{M}$.

2) $\mathcal{M}$ is part of response as shown in bottom part of Figure 2. Since there is no involvement of functional tools, we merge the *Planner* and *Executor* to alternatively plan and execute, in order to suit the complex strategy transition in practice. Notably, these complex strategy transition is pretty frequent such as $hint \rightarrow question \rightarrow hint$ in tutoring (Stasaski et al., 2020) and $EV \rightarrow RS \rightarrow EV$ in emotional support (Zheng et al., 2023). Due to the invoke of multiple identical strategies in a single turn, it is inferior to treat these same strategies at different steps independently once we separate the $Planner$ and $Executor$. Thus, the close interaction between the *Planner* and *Executor* allows for timely adjustments and responses to intermediate actions.

To conclude, we formulate the above two response logic of *Executor* as the following:

$$\text{resp} \leftarrow E^{w/\,func}(\mathcal{C}, \mathcal{K}; \mathcal{D}_e, Per_e), \quad \text{resp} \leftarrow combine\_middle^{w/o\,func}(\mathcal{M}), \tag{3}$$

where $\mathcal{D}_e$ and $Per_e$ represent the demonstration and persona of the *Executor* at the final response generation step, respectively. Both response clues are illustrated in Figure 2.

Overall, we identify our **TPE** framework that employs an extended *conceptual tool* set with the following distinctive attributes: 1) Specialized Focus: **TPE** framework distinguishes itself from the previous frameworks through its dedicated focus on dialogue systems, addressing the complexities of dynamic dialogue scenarios, and effectively orchestrating multiple resources and strategies for

---

[5]We find that adding examples besides name and description can not lead to improvement (§5.3).

planning. 2) Versatility: `TPE` exhibits versatility in its application, making it suitable for a wide range of multi-source or multi-strategy dialogues, as validated in our additional experiments. 3) Explainability: Within `TPE`, the reasoning processes and behaviors of the *Thinker*, *Planner*, and *Executor* can be precisely matched, ensuring transparency and explainability in the final responses. 4) Efficiency: the decoupling design (e.g., independent components without iterative processes) makes it more efficient by using less token consumption or computation cost.

## 4 EXPERIMENT

In this section, we conduct extensive experiments on two different dialogue response generation tasks necessitating the call of different sources or strategies, including FoCus (Jang et al., 2022) which involves two different sources of knowledge base: *Persona* and *Document*[6], CIMA (Stasaski et al., 2020) in which five strategies can be chosen to guide the student to solve the original problem: *Hint*, *Question*, *Correction*, *Confirmation*, and *Others*. We provide a detailed statistics analysis in the Appendix A.1. To evaluate the performance of our proposed method, we compare it with previous supervised methods and several strong unsupervised baselines based on LLM (§4.1), and the results demonstrate the effectiveness and efficiency of `TPE` (§4.2), and we additionally conduct experiments on PsyQA (Sun et al., 2021) in which seven (including others) psychological strategies can be selected by the professional counselors to generate counseling answers, aiming to validate the generalization capability of our proposed method (§4.3).

### 4.1 SET UP

**Evaluation Metrics:** We adopt different metrics to evaluate performance on the different datasets following previous works. Specifically, we use Avg.BLEU (a.k.a., the average of BLEU-1,2,3,4) (Papineni et al., 2002), F1 and Rouge.L (Lin, 2004) as three major metrics to evaluate the performance of the FoCus and PsyQA datasets following (Jang et al., 2022; Sun et al., 2021). Another two metrics besides F1 are used for the CIMA dataset are sacreBLEU (Post, 2018) and BERTScore (Zhang et al., 2020) following (Wang et al., 2023b). We also provide a human evaluation at the Appendix B.1.

**Baselines:** To ensure a comprehensive and equitable comparison, we have meticulously chosen a set of baseline models from both supervised and unsupervised settings. In the supervised category, we have selected two approaches: **GPT-2+PG+KG** and **BART+PG+KG** for FoCus (Jang et al., 2022), and **BART** as well as **mBART** for CIMA (Wang et al., 2023b); In the realm of unsupervised methods, we have focused on five distinct CoT approaches, namely **Vanilla CoT** (Wei et al., 2022), **ReAct** (Yao et al., 2023), **Chameleon** (Lu et al., 2023), **ReWOO** (Xu et al., 2023a), and **Cue-CoT** (Wang et al., 2023a). These methods have been specifically designed and validated to enhance the planning ability of large language models (LLMs) to call different functional tools and we re-implement these in the context of multi-source and multi-strategy dialogue response generation. We provide prompt details for each method in Appendix B.6.

**Implementation Details:** For supervised methods, we either directly copy the results or re-run the evaluation using the default setting from the original papers (Wang et al., 2023b; Jang et al., 2022). We utilize the same hyperparameters setting for all unsupervised baselines based on two SOTA LLMs: ChatGPT (gpt-3.5-turbo-0613) and GPT-4 (gpt-4-0613) (OpenAI, 2023). In detail, we set the temperature as 0 and the top p as 0.1 for inferences in all models, aiming to minimize the effects of randomness. We use the fixed examples in the demonstration with different formats to comply with different methods. Specifically, we use three examples for FoCus and CIMA and two examples for PsyQA datasets respectively. The number of demonstrations is chosen according to the unique characteristics of different datasets as explained in Appendix A.2. We use BM25 (Robertson et al., 2009) as the retriever and retrieve top-1 results from different sources for multi-source dialogue, and we also provide the experimental results of DPR (Karpukhin et al., 2020) as the retriever in the Appendix B.2.

---

[6]We use Document instead of Knowledge in the original paper (Jang et al., 2022) to reduce ambiguity.

Table 1: The performance of our proposed **TPE** method with previous supervised SOTA models and unsupervised various CoT methods. The supervised results of CIMA are directly coping from Wang et al. (2023b). We re-run the evaluation on the validation dataset of FoCus for all methods since the test dataset is not publicly available. The highest scores among models in each section are highlighted in blue , and the results of best performance are marked in **bold**.

| Method | FoCus | | | | CIMA | | | |
|---|---|---|---|---|---|---|---|---|
| | #Demos | Avg.B | F1 | Rouge.L | #Demos | sBLEU | F1 | BERTScore |
| BART | - | - | - | - | - | 8.67 | - | 70.80 |
| mBART | - | - | - | - | - | 11.90 | - | 73.00 |
| GPT-2 + PG + KG | - | 10.68 | 27.73 | 30.97 | - | - | - | - |
| BART + PG + KG | - | 11.34 | 28.87 | **31.11** | - | - | - | - |
| Supervised method (Above) △ | | | | | | | | |
| *Zero-shot ChatGPT* | | | | | | | | |
| CoT | - | - | - | - | 0 | 9.07 | 29.05 | 84.77 |
| *Few-shot ChatGPT* | | | | | | | | |
| CoT | 3 | 14.68 | 31.86 | 24.57 | 3 | 12.89 | 35.34 | 85.71 |
| ReAct | 3 | 19.55 | 33.74 | 27.05 | 3 | 13.43 | **53.13** | 84.73 |
| Cue-CoT | - | - | - | - | 3 | 10.86 | 41.42 | 85.39 |
| Chameleon | 3 | 14.98 | 31.70 | 24.47 | 3 | 11.17 | 29.77 | 84.24 |
| ReWOO | 3 | 21.05 | 34.45 | 28.77 | - | - | - | - |
| **TPE** | 3 | **23.47** | **36.95** | 30.64 | 3 | 14.46 | 36.42 | 85.76 |
| *Few-shot GPT-4* | | | | | | | | |
| CoT | 3 | 14.29 | 31.39 | 24.72 | 3 | 15.01 | 39.28 | **86.74** |
| ReWOO | 3 | 19.73 | 35.57 | 27.89 | - | - | - | - |
| **TPE** | 3 | 20.14 | 36.01 | 28.04 | 3 | **18.49** | 40.25 | 86.56 |

## 4.2 MAIN RESULT

Table 1 shows the main results. Generally, we observe compelling improvement for **TPE** over both supervised and unsupervised methods with the only exception at Rouge.L on FoCus. Notably, **TPE** consistently exhibits superior performance in 5 out of 6 evaluation metrics over all unsupervised baselines, irrespective of the employed type of LLMs. Specifically:

**Multi-source Dialogue.** It is worth noting CoT here is a special case of Chameleon whose default source order is fixed as *['Persona', 'Document']*[7]. By letting LLM plan the order of different sources, Chameleon achieves a higher Avg.B and comparable performance at F1 and Rouge.L. Furthermore, ReWOO achieves further improvement at all metrics by not only determining the source order but also the argument dependency (a.k.a, the query dependency). We conclude that the argument dependency between different sources is more important than simply the call order of different sources. In addition, we find that ReAct is capable of retrieving evidence from the same source if the retrieved results are not desired by predicting the same action repeatedly (Zhuang et al., 2023). This phenomenon elucidates the concurrent occurrence of improved performance and elevated costs, often referred to as reduced efficiency. We provide the efficiency analysis in Appendix A.3. With careful modeling of internal status and external sources of knowledge (such as call order of different sources, and argument dependency), **TPE** surpasses all CoT baselines, revealing its effectiveness and efficiency. Finally, we find that GPT-4 isn't particularly adept at handling multi-source tasks. We attribute this to the personal information stored in the persona source. Since they are not stored within the model's parameters, leading to a higher likelihood of generating incorrect or fabricated information, commonly referred to as *hallucinations* (Ji et al., 2023).

**Multi-strategy Dialogue.** The performance gap between different methods is relatively small compared with multi-source since there is no involvement of functional tools. In this way, we observe that CoT outperforms Chameleon a lot, and we empirically find that Chameleon easily determines the same strategy continuously. In addition, Cue-CoT achieves the second-best performance of baselines in F1 and BERTScore, revealing the effectiveness of considering the internal status exhibited during the conversation. Consistently, **TPE** demonstrates superior performance over these competitive baselines. We provide additional strategy analysis in § 5.2.

## 4.3 PSYCHOLOGICAL STRATEGIES

---

[7]We follow the implementation (Lu et al., 2023) to make a fair comparison.

We conduct additional experiments on psychological therapy situations where the responses require complex reasoning of seven professional psychological strategies Hill (2009); Sun et al. (2021). Table 2 shows the experimental results. Despite there being more complex strategies and longer responses in the dataset, **TPE** consistently delivers strong performance. We surprisingly find that the performance gap between Chameleon and others becomes bigger, especially when compared to the baseline vanilla CoT in a zero-shot setting. The reason for this performance disparity lies in how Chameleon operates. It generates sub-responses independently for different strategies, making it unable to handle the complex transition in multi-strategy dialogues, such as multiple calls of the same strategy *(Approval and Reassurance, Interpretation, Direct Guidance, Interpretation, Direct Guidance)* (Sun et al., 2021). It hurt the performance by ignoring the relationship between different strategies and the subtle differences when calling the same strategy at different steps in a single turn.

Table 2: The performance of TPE with previous strong CoT competitive on PsyQA dataset (Sun et al., 2021). We **bold** the best performance and underline the second-best.

| Method | PsyQA | | | |
|---|---|---|---|---|
| | #Demos | Avg.B | F1 | D-1 |
| *Zero-shot ChatGPT* | | | | |
| CoT | 0 | 8.14 | 17.37 | 20.70 |
| *Few-shot ChatGPT* | | | | |
| CoT | 2 | 15.62 | 32.66 | 43.12 |
| ReAct | 2 | 16.03 | 33.63 | 41.17 |
| Cue-CoT | 2 | 15.27 | 31.91 | 45.32 |
| Chameleon | 2 | 6.33 | 17.43 | 33.37 |
| **TPE** | 2 | 16.19 | 33.06 | 44.81 |
| *Few-shot GPT4* | | | | |
| CoT | 2 | 15.78 | 33.33 | **47.91** |
| **TPE** | 2 | **16.33** | **34.21** | 41.30 |

## 5 ANALYSIS

### 5.1 THE EFFECTS OF RETRIEVAL IN SOURCE PLANNING

In the context of generating responses, aside from the order in which sources are accessed, the accuracy of retrieving relevant information is crucial. Two main factors that significantly affect this accuracy are *the semantic gap between the query with document* and *the number of results retrieved*.

Instead of solely using the dialogue context as the query (Yu et al., 2021; Xu et al., 2022), we use the internal status generated by the *Thinker* to enrich the semantic information of the query, achieving better performance as shown in the right part of Figure 3. Notably, when we remove the internal status from the process, we observe a drop in all metrics by approximately 3%. In addition, we also investigate the effects of the number of retrieved results by increasing it from the original 1 to 4. We find that it usually leads to better performance, primarily because LLMs can effectively capture relevant information from the lengthy retrieved results.

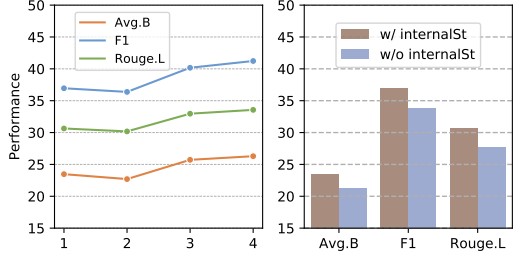

Figure 3: **Left**: The effects of different numbers of retrieved results; **Right**: The effects of using internal status to enrich the query.

It's worth noting that the absence of conflicting knowledge in the dataset also contributes to this positive effect. However, it also brings new problems. As the number of retrieved results increases, the input length also increases, resulting in higher inference costs. We were unable to set the number to 5 due to exceeding the input limit. Further details regarding inference costs and the count of correctly retrieved evidence are provided in the Appendix B.3.

### 5.2 PERFORMANCE OF STRATEGY PLANNING

In order to directly evaluate the planning capability of LLMs with the help of **TPE**, we conduct an analysis by comparing the strategies generated by the ReAct and **TPE** with different backbones (a.k.a, ChatGPT, and GPT-4)[8], and the ground-truth strategies in the dataset. Figure 4 shows the strategy distribution under these three situations. There are some interesting observations: 1) ReAct, as an observation-dependent counterpart, tends to get stuck in suboptimal planning due to an overemphasis on the Hint strategy; 2) The most used strategy of **TPE** is the correction. Since stu-

---

[8]We choose ReAct since it performs best out of all CoT baselines and we do not report ReAct using GPT-4 due to the computation cost.

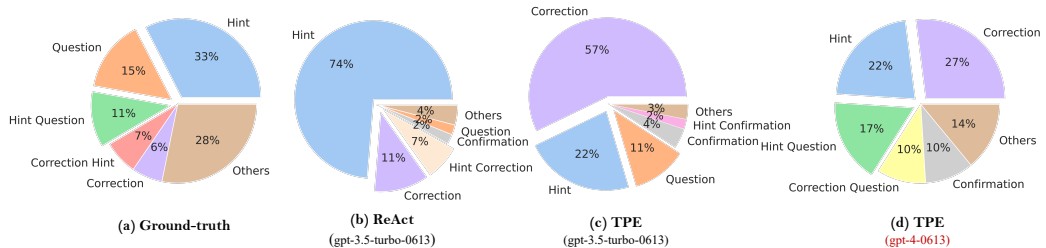

Figure 4: The strategy distribution shifts under different situations: (a) Ground-truth; (b) ReAct with ChatGPT; (c) **TPE** with ChatGPT; and (d) **TPE** with GPT-4. It is worth noting here the **Others** is not the same as the *others* strategy defined before but stands for all others strategies combinations such as Confirmation, Question Hint Correction, and so on.

dents often make errors or harbor misconceptions when seeking assistance, these LLMs typically opt for a direct correction strategy rather than offering subtle hints to guide the student once they capture the internal status exhibited during dialogue. Nevertheless, it is satisfying to observe that this prevalent phenomenon is mitigated when employing a more advanced model, namely GPT-4 (57% → 27%); 3) LLMs are capable of employing new and combined strategies with only definitions of strategies, such as *confirmation* and *hint confirmation* in TPE (ChatGPT) and *correction question* in TPE (GPT-4). Notably, TPE (GPT-4) demonstrates a significantly higher proficiency in employing combined strategies compared to TPE (ChatGPT) (27% v.s. 2%); 4) Demonstrations are essential for teaching LLMs when and how to employ specific strategies effectively. These strategies, except for correction, are employed by LLMs at rates of 33% in TPE (ChatGPT) and 39% in TPE (GPT-4), as per the demonstrated examples.

### 5.3 WHAT MATTERS IN IN-CONTEXT LEARNING

There are several elements that significantly contribute to in-context learning, including strategy description, strategy examples, and demonstrations. Thus we provide a comprehensive ablation study in this section by investigating the effects of different components in the context of multi-strategy dialogues[9]. Table 3 shows the results. We observe that both adding strategy examples and

Table 3: The effects of different components for in-context learning in CIMA dataset (Stasaski et al., 2020). We **bold** the best performance and underline the second-best.

| Method | #Demos | sBLEU | F1 | BERTScore |
|---|---|---|---|---|
| TBD | 3 | 14.46 | 36.42 | 85.76 |
| + Strategy examples | 3 | 11.34 | 31.50 | 85.04 |
| - Strategy desc | 3 | 12.68 | 35.19 | 85.05 |
| + Demos (Hint) | 4 | 13.89 | 38.54 | **86.05** |
| + Demos (Correction) | 4 | 14.19 | **38.83** | 85.94 |
| - Demos (Hint) | 2 | **16.33** | 35.96 | 85.33 |

removing strategy descriptions lead to deteriorated performance. Interestingly, the worst performance is observed when strategy examples are added. Our analysis of the strategy distribution, depicted in Appendix B.5, indicates that removing descriptions does not significantly alter the strategy planning distribution. This suggests that the strategy names themselves may convey self-explanatory semantic signals. However, the inclusion of examples appears to confuse LLMs, leading them to utilize the *others* strategy more frequently. Besides that, we noted that when LLMs determine the same strategy, they tend to follow a similar synthetic structure following the provided examples.

## 6 CONCLUSION

In this paper, we explore the planning capability of LLMs further by introducing *conceptual tools*, especially in the context of the dialogue system. Furthermore, we present a novel multi-persona framework: **TPE** to solve multi-source and multi-strategy dialogue tasks collaboratively. By achieving superior performance over existing methods on three datasets (FoCus, CIMA, and PsyQA), **TPE** showcases its potential for many applications that necessities the involvement of *conceptual tools*. We have reserved the exploration of more complex combinations of *functional* and *conceptual* tools for future work.

---

[9]We provide multi-source analysis in Appendix B.4. More analysis can be found in Appendix B.5.

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

| Sources | Definitions | Examples |
|---|---|---|
| PERSONA | This knowledge base stores personal preferences or relevant personal details about the user. It takes in the query and returns a related user persona that assists in addressing the user's current question or input. | I am interested in History. I would like to visit the Nazareth House again. |
| DOCUMENT | This knowledge base stores background knowledge from Wikipedia as the hint for the given dialogue. Normally, we consider using DOCUMENT when the background knowledge is required and helpful to guide the response generation. | Shortly after acquiring the property, the museum underwent a two-year renovation beginning in 1989. Nazareth House is a heritage-listed benevolent institution at Australia. |

Table 5: The definition of different sources in FoCus (Jang et al., 2022). We write the definition according to the natural characteristics of different sources.

**Tutor Actions**

| Action Label | Description | Example |
|---|---|---|
| Hint | The tutor provides knowledge to the student via a hint. | "Here's a hint - "tree" is "l'albero" because l' ("the") is prepended to the following word when it begins with a vowel." |
| Open-Ended Question | The tutor asks a question of the student, which can attempt to determine a student's understanding or continue the conversation. | "Are you sure you have all the words in the right order?" |
| Correction | The tutor corrects a mistake or addresses a misconception a student has. | "Very close. Everything is correct, expect you flipped 'viola' and 'coniglio'." |
| Confirmation | The tutor confirms a student's answer or understanding is correct. | "Great! Now say the whole sentence, starting with the dog..." |
| Other | We allow tutors to define a category if they do not believe their response fits into the predefined categories. | "Correct! Although try to think of the complete word as 'la scatola.' I find that the easiest way to remember what gender everything is - I just think of the 'the' as part of the noun." |

Table 6: The definition and example of different strategies in CIMA (Stasaski et al., 2020).

## A  DATASET ANALYSIS

In this section, we introduce more details about the used datasets and explain how the number of demonstrations is determined.

### A.1  DATASET STATISTICS

We sample 200 samples from each dataset to conduct our experiments, and we provide data statistics at Table 4. As we mentioned in our main paper, there are two sources (Persona and Document) in the FoCus dataset, five strategies in CIMA and seven strategies in PsyQA. We provide definitions and examples of these sources and strategies in Table 5, 6, and 7, respectively. It is worth noting that there are **five** candidates in the persona source and **ten** candidates in the document source in the FoCus dataset.

### A.2  THE SELECTION OF DEMONSTRATIONS

We choose different numbers of demonstrations according to differences across datasets.

**FoCus.** Specifically, there are three situations in the datasets: 1) the response only requires the document source without requiring the persona source; 2) the response needs the persona source first and then the document; 3) the re-

Table 4: The data statistics of three datasets

| Datasets | FoCus | CIMA | PsyQA |
|---|---|---|---|
| Testing Samples | 200 | 200 | 200 |
| Min number of sources/strategies | 2 | 1 | 1 |
| Max number of sources/strategies | 2 | 5 | 7 |
| Avg number of sources/strategies | 2 | 1.5 | 3.8 |
| Max length of resp | 819 | 262 | 1842 |

| Strategies | Definitions | Examples | Lexical Features |
|---|---|---|---|
| Information | Supply information in the form of data, facts, opinions and resources. | 心理学中有个关于"初恋"的效应, 叫"蔡格尼克记忆效应"。 *There is a psychological effect on first love, called Zeigarnic effect.* | 指/refer to (3), 心理学/psychology (3), 心理学家/psychologist (3), 研究/survey (3), 效应/effect (3) |
| Direct Guidance | Provide suggestions, directives, instructions, or advice about what the help-seeker should do to change. | 如果觉得难以改变, 可以寻求靠谱的心理咨询师的帮助。 *If you find it hard to change, you can seek help from a trusted counselor.* | 建议/advice (9), 尝试/try (8), 学会/learn (6), 找/find (5), 沟通/communicate (5) |
| Approval and Reassurance | Emotional support, reassurance, encouragement and reinforcement. | 给你温暖的抱抱呀! *Let me give you a warm hug!* | 抱抱/hug (15), 温暖/warm (8), 世界/world (7), 祝/wish (6), 心疼/care (5) |
| Restatement | A simple repeating or rephrasing of the content or meaning of the question, usually in a more concrete and clear way. | 您感觉自己产生了暴虐心理。 *You feel like you are becoming violent.* | 描述/description (4), 了解/understand (3), 感觉/feel (3), 说/say (3), 提到/mention (2) |
| Interpretation | Go beyond what the help-seeker has overtly stated or recognized and give a new meaning, reason or explanation. | 我想你是很爱很爱妈妈的。 *I think you love your mom very much.* | 会/will (6), 人/people (5), 是/be (4), 每个/every (3), 知道/know (3) |
| Self-disclosure | Reveal something personal about the helper's non-immediate experiences or feelings. | 这个问题勾起了我类似的回忆。 *This question brings back to me some similar memories.* | 我/I (2), 爷爷/grandpa (2), 大学/college (2), 外婆/grandma (1), 供养/raise (1) |

Table 7: The definition and example of different strategies in PsyQA (Sun et al., 2021), together with the lexical features of the strategies. The rightmost column displays the top 5 words associated with each strategy.

sponse needs the document source first and then the persona source. The first situation is well-explained in the original paper (persona selection is a multi-label classification task and document selection is a multi-class classification task) and also validated by us in the dataset. We provide a detailed explanation for the latter two situations in Figure 5. Thus we use three demonstrations to illustrate these three situations.

**CIMA.** We choose the **top three** most used strategy transitions: *Hint*, *Question*, and *Hint Question* as demonstrations, we also provide an analysis of different demonstration selection strategies in the subsequent section.

**PsyQA.** We first choose one demonstration with the strategy transition as (*Approval and Reassurance, Interpretation, Direct Guidance, Interpretation, Direct Guidance*), since this is a common and typical flow as reported by Sun et al. (2021). We then carefully choose another demonstration in order to increase the diversity of the strategies.

## A.3 EFFICIENCY ANALYSIS

Inspired by recent work (Xu et al., 2023a) which uses the number of used tokens as an indicator to evaluate the efficiency of different CoT methods, we here adopt a similar idea by presenting the money cost for different methods with two major backbones. It is more straightforward and direct since the LLMs used here are charged by the tokens with the same rates[10]. Table 8 shows the whole cost during our experiments for each method under the few-shot setting. Notably, ReAct costs the highest money especially when the strategy transition becomes more complex and the response becomes longer (PsyQA). By decoupling the compositional reasoning into multi-persona components, **TPE** incurs comparable costs to previous methods that

Table 8: The money cost by calling APIs from OpenAI using different methods. We bold the highest cost and underline the second-highest.

| Method | FoCus | CIMA | PsyQA |
|---|---|---|---|
| *Few-shot ChatGPT* | | | |
| CoT | 0.37 | 0.10 | 0.60 |
| ReAct | **2.31** | **0.74** | **8.11** |
| Cue-CoT | - | 0.27 | 1.21 |
| Chameleon | 0.58 | 0.23 | 1.94 |
| ReWOO | 0.42 | - | - |
| **TPE** (ChatGPT) | 0.86 | 0.25 | 1.54 |

[10]In general, 0.002 USD / 1k tokens for GPT-3.5-Turbo and 0.03 USD / 1k tokens for GPT-4.

do not rely on observation-dependent factors, yet it attains superior performance and better explainability.

## B  EXPERIMENTAL ANALYSIS

### B.1  HUMAN EVALUATION

We randomly sample 100 dialogue contexts for FoCus and CIMA, accompanied by responses generated using various methods. We then ask three well-educated annotators to assign a quality score ranging from 1 to 5 for each response, without revealing the methods employed in generating these responses. Specifically, we asked them to assign scores considering the different settings of each dataset. For CIMA, we encourage them to prioritize the level of inspiration the responses provide to the students rather than simply providing solutions. In contrast, we request they pay more attention to the source knowledge to ensure both the correctness and informativeness of responses for FoCus. Table 9 shows the final results

Table 9: The human evaluation results. We bold the highest performance.

| Methods | FoCus | CIMA |
|---|---|---|
| CoT | 2.56 | 3.30 |
| ReAct | 2.76 | 2.80 |
| Cue-CoT | - | 3.32 |
| Chameleon | 2.72 | 3.20 |
| ReWOO | 3.14 | - |
| **TPE** | **3.28** | **3.66** |

and the inter-agreement is about 75%. It is obvious that **TPE** outperforms other baselines and the gap between CIMA is relatively larger, revealing the effectiveness of our method on the complex strategy transition.

### B.2  DIFFERENT TYPES OF RETRIEVER IN MULTI-SOURCE

We additionally utilize dense vector retriever – DPR (Karpukhin et al., 2020), which is initialized with the 12-layer `bert-base-uncased` model. We firstly directly use the pre-trained model without in-domain finetuning as the retriever and calculate the cosine similarity using the [CLS] representation. Furthermore, we finetune DPR using FoCus dataset by regarding *(context, used_persona / document)* as the positive and *(context, unrelated_persona / document)* as

Table 10: The performance of `TPE` with different types of retriever on FoCus.

| Retriever | Avg.B | F1 | Rouge.L |
|---|---|---|---|
| BM25 | 23.47 | 36.95 | 30.64 |
| DPR w/o finetune | 17.63 | 30.12 | 23.95 |
| DPR w finetune | **28.00** | **43.14** | **35.18** |

the negative. We set epochs as 5 and max sequence length as 512, and mainly follow the scripts: `https://github.com/Alibaba-NLP/Multi-CPR/tree/main/retrieval` for other parameters. Table 10 shows the results of different types of retrievers. We found the DPR without finetuning can not achieve on-par performance with BM25, and it brings about 5% improvement after finetuning compared with spare retriever – BM25.

### B.3  RETRIEVAL COST AND ACCURACY

We directly evaluate the performance of retriever – BM25 used in main experiments when setting different numbers of retrieved results. Table 11 shows the results. Generally, as the number of retrieved results increases, the number of correct personas or documents also increases, with the increasing cost due to more tokens consumption. Furthermore, we found the number of correct persons is much lower than the number of correct documents. We attribute this phenomenon to the larger semantic gap between the persona and the dialogue con-

Table 11: The number of correct personas and documents with the money cost under the different numbers of retrieved results with ChatGPT as the backbone.

| Number | 1 | 2 | 3 | 4 |
|---|---|---|---|---|
| # correct persona | 19 | 45 | 64 | 85 |
| # correct document | 237 | 287 | 291 | 323 |
| # money cost | 0.86 | 0.92 | 0.94 | 0.99 |

text. Without the internal status to enrich the semantic information of dialogue context, the number of correct personas is further dropped to 11.

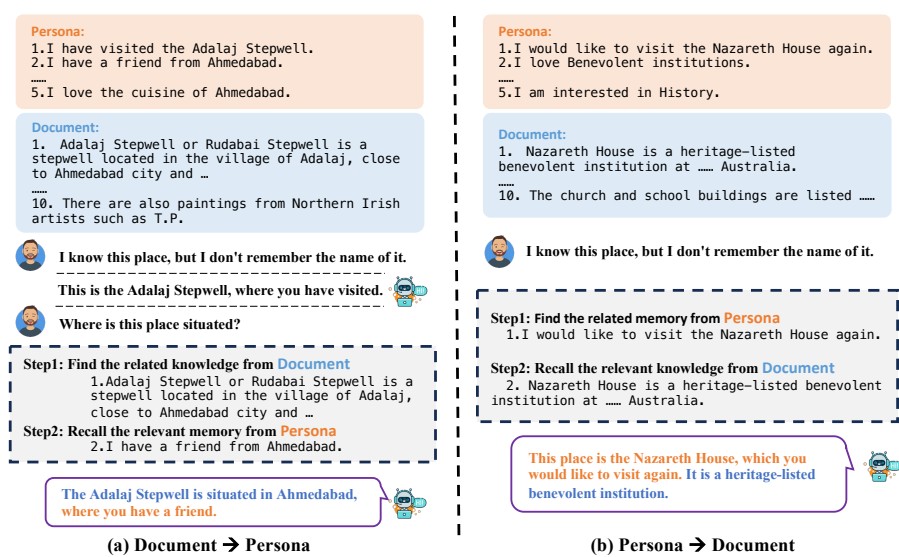

Figure 5: Illustration of different orders of sources planning in FoCus dataset.

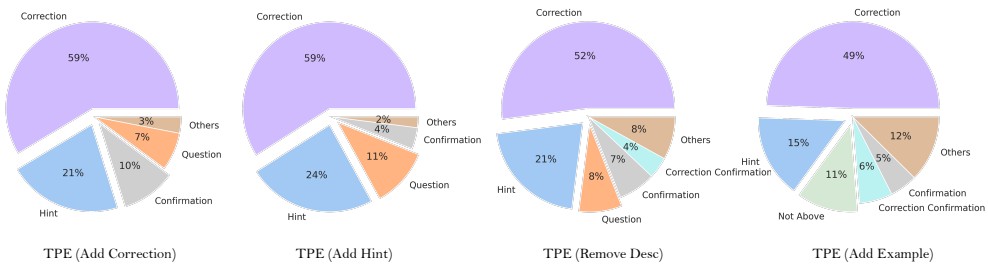

Figure 6: The strategy distribution shifts with different demonstrations with ChatGPT as the backbone. It is worth noting that Not Above in the last sub-figure means the *others* strategy defined in the dataset.

## B.4    IN-CONTEXT IN MULTI-SOURCE

Similar to the analysis conducted in Section 5.3, we investigate the effects of source examples and description for the final performance of **TPE**. Table 12 shows the results. We found that the trend is consistent with the CIMA dataset. Adding source examples and removing descriptions both lead to worse performance.

## B.5    STRATEGY PLANNING WITH DIFFERENT DEMONSTRATION

The impact of introducing or removing various demonstrations on final performance is intricate, yielding both improved and diminished metrics depending on the specific dataset and the order of demonstrations (Lu et al., 2022). In general, the addition of demonstrations tends to increase the adoption of corresponding strategies, such as correction and hints shown in the left bottom of Figure 6.

Table 12: The effects of different components for in-context learning in FoCus dataset (Jang et al., 2022). We **bold** the best performance and underline the second-best.

| Method | #Demos | Avg.B | F1 | Rouge.L |
|---|---|---|---|---|
| **TPE** | 3 | **23.47** | **36.95** | **30.64** |
| + Source examples | 3 | 23.27 | 36.30 | 29.75 |
| - Source desc | 3 | 23.43 | 36.38 | 30.12 |

### B.6  PROMPT DETAILS

Due to the unique characteristics of different CoT methods, there are differences between the content and format of prompts. In addition, we want to emphasize we utilize dialogue context as part of the query for all CoT methods that consider the query dependency between different sources, including ReWOO, ReAct, and our `TPE`.

**Chameleon:** Table 13  14 for FoCus and CIMA, respectively. Since this method requires independent module demonstrations, we additionally provide different demonstrations for each strategy in CIMA as shown in Table 15.

**ReAct:** Table 16  17 for FoCus and CIMA, respectively.

**ReWOO:** Table 18 for FoCus. It's important to highlight that as the number of strategies increases, the number of evidence (#E(number)) can become uncontrollable, making it impractical to present the results of ReWOO on the CIMA and PsyQA datasets.

**CueCoT:** There are two steps in CueCoT (Wang et al., 2023a), in which the first step is used to infer the user status, and the second step generates the final response. We follow this consecutive setting to first prompt LLMs to infer the status and then generate the response respectively, with the same examples as demonstrations. Table 19 presents the prompt details, we merge two steps together to save space.

**TPE:** Table 20,  21 for FoCus and CIMA, respectively.  We provide definitions of each input of different roles as shown in Figure 2.

Table 13: Prompts and exemplars used in Chameleon for the FoCus dataset

| CHAMELEON |
|---|
| You need to act as a policy model, that given a dialogue and a modular set, determines the sequence of modules that can be executed sequentially can solve the question. |

The modules are defined as follows:
- *Persona_Retrieval*: This module retrieves personal preferences or relevant personal details about the user. It takes in the query and returns a related user persona that assists in addressing the user's current question.
- *Knowledge_Retrieval*: This module retrieves background knowledge from Wikipedia as the hint for the given dialogue. Normally, we consider using "Knowledge_Retrieval" when the background knowledge is helpful to guide the solution.
- *Answer_Generator*: This module extracts the final answer in a short form from the solution or execution result. This module normally is the last module in the prediction pipeline.

Below are some examples that map the dialogue to the modules.

**Dialogue**: USER: What is the geography of this place?  SYSTEM: The Arctic Cordillera is geographically diverse, and much of Ellesmere Island is covered by the Arctic Cordillera, making it the most mountainous in the Canadian Arctic Archipelago. You would love this place since you are interested in geography.  USER: What is the overview of this area?
**Modules**: ["Knowledge_Retrieval", "Answer_Generator"]

**Dialogue**: USER: I know this place, but I don't remember the name of this place.
**Modules**: ["Persona_Retrieval", "Knowledge_Retrieval", "Answer_Generator"]

**Dialogue**: USER: Wow, this is amazing! What is this?  SYSTEM: This is The Arctic Cordillera located in Canada, which you want to visit.  USER: What is the place known for?
**Modules**: ["Knowledge_Retrieval", "Persona_Retrieval", "Answer_Generator"]

Table 14: Prompts and exemplars used in Chameleon planner for the CIMA dataset

CHAMELEON

You are a teacher who helps a student translate a phrase from English to Italian. Given a dialogue and a strategy set, determine the sequence of strategies that can be executed sequentially to guide the student.
The strategies are defined as follows:
- *Hint*: The teacher provides knowledge to the student via a hint.
- *Question*: The teacher asks a question of the student, which can attempt to determine a student's understanding or continue the conversation.
- *Correction*: The teacher corrects a mistake or addresses a misconception a student has.
- *Confirmation*: The teacher confirms a student's answer or understanding is correct.
- *Others*: Refers to any strategy or approach that does not fall within the predefined categories.

Below are some examples that map the dialogue to the strategies.

**Dialogue**: Teacher: "Is Inside Of The" is "e dentro la". Please try to fill in the blank in Italian.   Student: How do you say blue box in Italian?   Teacher: Prepositional phrases separate the two noun phrases.   Student: Is it e dentro la box blu?
**Strategies**: ['Hint', 'Question']

**Dialogue**: Teacher: Green is verde. Please try to fill in the blank in Italian.   Student: what is the word for green?
**Strategies**: ['Hint']

**Dialogue**: Teacher: 'Is Behind The' is 'e dietro il'. Please try to fill in the blank in Italian.   Student: what is blue in italian?   Teacher: Can you give me your best guess?   Student: blueo   Teacher: Remember that 'is behind the' is 'e dietro il'   Student: e dietro il blueo cato   Teacher: Hmm... 'is behind the' is 'e dietro il'   Student: e dietro il   Teacher: Hmm... 'cat' is 'gatto'   Student: e dietro il gatto
**Strategies**: ['Question']

Table 15: Prompts and exemplars used in Chameleon strategy components for the CIMA dataset

| CHAMELEON STRATEGY COMPONENTS |
| --- |

Hint **Dialogue**: Teacher: "Is Inside Of The" is "e dentro la". Please try to fill in the blank in Italian. Student: How do you say blue box in Italian? Teacher: Prepositional phrases separate the two noun phrases. Student: Is it e dentro la box blu?
**Hint**: box is scatola.
**Dialogue**: Teacher: Green is verde. Please try to fill in the blank in Italian. Student: what is the word for green?
**Hint**: la pianta e dentro la scatola verdeverde
**Dialogue**: Teacher: Please try to fill in the blank in Italian. Student: How do you say in front of? Teacher: Why don't you try filling in what you can. Student: il coniglio e front il tree verde
**Hint**: 'in front of' is 'e di fronte'.

Question **Dialogue**: Teacher: "Is Inside Of The" is "e dentro la". Please try to fill in the blank in Italian. Student: How do you say blue box in Italian? Teacher: Prepositional phrases separate the two noun phrases. Student: Is it e dentro la box blu?
**Question**: Do you remember how to say the plant?
**Dialogue**: Teacher: 'Is Behind The' is 'e dietro il'. Please try to fill in the blank in Italian. Student: what is blue in italian? Teacher: Can you give me your best guess? Student: blueo Teacher: Remember that 'is behind the' is 'e dietro il' Student: e dietro il blueo cato Teacher: Hmm... 'is behind the' is 'e dietro il' Student: e dietro il Teacher: Hmm... 'cat' is 'gatto' Student: e dietro il gatto
**Question**: great but what color is the cat? and who is behind the cat, how do you say bunny?
**Dialogue**: Teacher: Please try to fill in the blank in Italian. Student: How do you say in front of? Teacher: Why don't you try filling in what you can. Student: il coniglio e front il tree verde
**Question**: Do you know the word for tree in Italian?

Correction **Dialogue**: Teacher: Please try to fill in the blank in Italian. Student: how do you say bed Teacher: Okay, I'll give you a hint. 'bed' is 'letto' Student: il cane es dieplo letto?
**Correction**: Remember, 'behind' is 'e dietro il' in Italian.
**Dialogue**: Teacher: Please try to fill in the blank in Italian. Student: e si cima ell yellow table
**Correction**: 'Yellow Table' is incorrect.
**Dialogue**: Teacher: 'Is Behind The' is 'e dietro il'. Please try to fill in the blank in Italian. Student: what is blue in italian? Teacher: Can you give me your best guess? Student: blueo
**Correction**: no, it's blu.

Confirmation **Dialogue**: Teacher: Please try to fill in the blank in Italian. Student: how do you say bed Teacher: Okay, I'll give you a hint. 'bed' is 'letto' Student: il cane es dieplo letto?
**Confirmation**: correct
**Dialogue**: Teacher: 'Is Under The' is 'e sotto il'. Please try to fill in the blank in Italian. Student: How do you say bed in Italian? Teacher: il ('the') is used for when the following word (letto) is masculine. Words in Italian have a gender associated with them (either masculine or feminine), even when the word is an object, concepts, or abstract ideas. Student: So, letto means bed? Teacher: Remember that 'bed' is 'letto' Student: Ok, I think I have it then,
**Confirmation**: Great! Let's go for it
**Dialogue**: Teacher: Please try to fill in the blank in Italian. Student: il gatto è vicino all'albero verde
**Confirmation**: Very good, that's correct!

Others **Dialogue**: Teacher: 'Is Behind The' is 'e dietro la'. Please try to fill in the blank in Italian. Student: what green is in Italian again? Teacher: OK, 'green' is 'verde' Student: Right! What is behind in Italian? Teacher: Well, 'is behind the' is 'e dietro la' Student: Oh yeah! So the first part is 'la borsa e dietro la verde'. What is box again? Teacher: Remember that 'box' is 'scatola' Student: la borsa e dietri la verde scatola Teacher: Prepositional phrases separate the two noun phrases. Student: Can you elaborate?
**Others**: 'E dietro la' is a prepositional phrase which comes between the two noun phrases, 'la borsa' and 'scatola verde.'
**Dialogue**: Teacher: 'Bunny' is 'coniglio'. Please try to fill in the blank in Italian. Student: e fronte il greene coniglio Teacher: Well, 'is in front of the' is 'e di fronte al' Student: e di fronte al greenee coniglio
**Others**: 'Greenee'? Oh, no. I don't think so!
**Dialogue**: Teacher: Please try to fill in the blank in Italian. Student: how do you say box? Teacher: Remember that 'box' is 'scatola' Student: e dentro de la scatola amarilla
**Others**: You got most of it.

Table 16: Prompts and exemplars used in ReAct for the FoCus dataset

| REACT |
| --- |

Generate a final response for the given dialogue with interleaving **Thought**, **Action**, **Observation** steps. **Thought** can reason about the current situation, and **Action** can be three types:
(1) *Persona[sentence]*, which retrieves personal preferences or relevant personal details about the user. It takes in the query and returns a related user persona that assists in addressing the user's current question.
(2) *Knowledge[sentence]*, which retrieves background knowledge from Wikipedia as the hint for the given dialogue. Normally, we consider using "Knowledge" when the background knowledge is helpful to guide the solution.
(3) *Finish[response]*, which returns the response and finishes the task.
Here are some examples.
**Dialogue**: USER: What is the geography of this place? SYSTEM: The Arctic Cordillera is geographically diverse, and much of Ellesmere Island is covered by the Arctic Cordillera, making it the most mountainous in the Canadian Arctic Archipelago. You would love this place since you are interested in geography. USER: What is the overview of this area?
**Thought**: I need to search for more information about the Arctic Cordillera.
**Action**: Knowledge[The Arctic Cordillera]
**Observation**: The range is mainly located in Nunavut but expands southeast into the northernmost tip of Labrador and northeastern Quebec. The system is split into a series of ranges, with mountains reaching heights of over 2,000 m (6,562 ft) while the highest is Barbeau Peak on Ellesmere Island at 2,616 m (8,583 ft), which is the highest point in eastern North America.
**Thought**: The retrieved knowledge is related and it can be used to generate the response.
**Action**: Finish[The range is mostly located in Nunavut but extends southeast into the northernmost tip of Labrador and northeastern Quebec. The system is divided into a series of ranges, with mountains reaching heights of more than 2,000 m (6,562 ft). The highest is Barbeau Peak on Ellesmere Island at 2,616 m (8,583 ft), which is the highest point in eastern North America.]

**Dialogue**: USER: I know this place, but I don't remember the name of this place.
**Thought**: I need to search for personal memory about the place.
**Action**: Persona[context]
**Observation**: I like living in a city. I don't hope to ever visit New Zealand.
**Thought**: The USER like living in a city and does not hopt to visit New Zealand. I need to know information about New Zealand.
**Action**: Knowledge[New Zealand]
**Observation**: Newton is a small suburb of Auckland City, New Zealand, under the local governance of the Auckland Council.
**Thought**: The persona and knowledge both are helpful to generate the final response. I can use them together.
**Action**: Finish[It's called Newton and it is a small suburb of Auckland City in New Zealand, a neighborhood where you could comfortably live in if you were to travel to New Zealand, but you don't seem to hope for it.]

**Dialogue**: USER: Wow, this is amazing! What is this? SYSTEM: This is The Arctic Cordillera located in Canada, which you want to visit. USER: What is the place known for?
**Thought**: I need to search for more information about the place.
**Action**: Knowledge[The Arctic Cordillera]
**Observation**: The Arctic Cordillera is a terrestrial ecozone in northern Canada characterized by a vast, deeply dissected chain of mountain ranges extending along the northeastern flank of the Canadian Arctic Archipelago from Ellesmere Island to the northeasternmost part of the Labrador Peninsula in northern Labrador and northern Quebec, Canada.
**Thought**: Maybe the USER is interested in some special features of this place. I need to figure it out by searching the USER preferences.
**Action**: PERSONA[The Arctic Cordiller]
**Observation**: I am interested in ecozone.
**Thought**: The USER is intesested in ecozone. The persona and knowledge both are helpful in generating the final response. I can use them together.
**Action**: Finish[As you are interested in ecozone, you should know that the Arctic Cordillera is known as a terrestrial ecozone in northern Canada characterized by a vast, deeply dissected chain of mountain ranges extending along the northeastern flank of the Canadian Arctic Archipelago from Ellesmere Island to the northeasternmost part of the Labrador Peninsula in northern Labrador and northern Quebec, Canada.]

**Dialogue**: {dialogue} {agent_scratchpad}

Table 17: Prompts and exemplars used in ReAct for the CIMA dataset

| REACT |
|---|
| You are a teacher who helps a student translate a phrase from English to Italian. You need to adopt strategies that conform with some educational conversational norms, such as providing hints versus asking questions in appropriate contexts according to the student's status. Let's think step by step. |

(1) *Hint*: The teacher provides knowledge to the student via a hint.
(2) *Question*: The teacher asks a question of the student, which can attempt to determine a student's understanding or continue the conversation.
(3) *Correction*: The teacher corrects a mistake or addresses a misconception a student has.
(4) *Confirmation*: The teacher confirms a student's answer or understanding is correct.
(5) *Others*: Refers to any strategy or approach that does not fall within the predefined categories.
(8) *Response*: Combines all observations, and forms the final response.
Here are some examples.
**Dialogue**: Teacher: "Is Inside Of The" is "e dentro la". Please try to fill in the blank in Italian. Student: How do you say blue box in Italian? Teacher: Prepositional phrases separate the two noun phrases. Student: Is it e dentro la box blu?
**Thought**: I need to provide a hint
**Action**: Hint
**Observation**: box is scatola.
**Thought**: Then I need to ask a question to determine a student's understanding
**Action**: Question
**Observation**: Do you remember how to say the plant?
**Thought**: Now I combine them all into the final response
**Action**: Response
**Dialogue**: Teacher: Green is verde. Please try to fill in the blank in Italian. Student: what is the word for green?
**Thought**: I need to provide a hint
**Action**: Hint
**Observation**: la pianta e dentro la scatola verdeverde
**Thought**: Now I combine them all into the final response
**Action**: Response
**Dialogue**: Teacher: 'Is Behind The' is 'e dietro il'. Please try to fill in the blank in Italian. Student: what is blue in italian? Teacher: Can you give me your best guess? Student: blueo Teacher: Remember that 'is behind the' is 'e dietro il' Student: e dietro il blueo cato Teacher: Hmm... 'is behind the' is 'e dietro il' Student: e dietro il Teacher: Hmm... 'cat' is 'gatto' Student: e dietro il gatto
**Thought**: I need to ask a question to determine a student's understanding
**Action**: Question
**Observation**: great but what color is the cat? and who is behind the cat, how do you say bunny?
**Thought**: Now I combine them all into the final response
**Action**: Response
**Dialogue**: {context}
{agent_scratchpad}

Table 18: Prompts and exemplars used in ReWOO for the FoCus dataset

| REWOO |
| --- |
| Dialogue: USER: What is the geography of this place?  SYSTEM: The Arctic Cordillera is geographically diverse, and much of Ellesmere Island is covered by the Arctic Cordillera, making it the most mountainous in the Canadian Arctic Archipelago.  You would love this place since you are interested in geography.  USER: What is the overview of this area?
**–PLANNER–**
Plan: Search for more information about the Arctic Cordillera.
#E1 = KNOWLEDGE[The Arctic Cordillera]

Dialogue: USER: I know this place, but I don't remember the name of this place.
**–PLANNER–**
Plan: Search for personal memories about the place.
#E1 = PERSONA[context]
Plan: Search for more information about the place.
#E2 = KNOWLEDGE[E1]

Dialogue:  USER: Wow, this is amazing! What is this?  SYSTEM: This is The Arctic Cordillera located in Canada, which you want to visit.  USER: What is the place known for?
**–PLANNER–**
Plan: Search for more information about the place.
#E1 = KNOWLEDGE[The Arctic Cordillera]
Plan: Search for personal preferences about the place.
#E2 = PERSONA[E1] |

Table 19: Prompts and exemplars used in Cue-CoT for the CIMA dataset

| 1ST STEP IN CUE-COT |
| --- |
| You are a teacher who helps a student translate a phrase from English to Italian. You need to infer the student's confusion at the current step and what is the correct direction for the student to the final solution. **(1st step)** You need to adopt strategies that conform with some educational conversational norms, such as providing hints versus asking questions in appropriate contexts according to the student's status. Let's think step by step. **(2nd step)**
Here are some examples.
**Dialogue**:  Teacher:  "Is Inside Of The" is "e dentro la".  Please try to fill in the blank in Italian.  Student: How do you say blue box in Italian?  Teacher: Prepositional phrases separate the two noun phrases.  Student: Is it e dentro la box blu?
**Status**: The student in the current step appears to be somewhat confused about the structure of the translation. While they have made an attempt by saying, "Is it e dentro la box blu?" there is a slight confusion in the word order and article usage. The correct direction for the student is to adjust the word order to "La box blu è dentro" to maintain the correct Italian sentence structure.
**Response:** box is scatola. Do you remember how to say the plant?

**Dialogue**: Teacher: Green is verde. Please try to fill in the blank in Italian.  Student: what is the word for green?
**Status**: The student in the current step appears to be confused about the Italian word for "green". The correct direction for the student is to use the information provided by the teacher and fill in the blank with "verde," as instructed, to complete the translation.
**Response:** la pianta e dentro la scatola verdeverde

**Dialogue**: Teacher: 'Is Behind The' is 'e dietro il'. Please try to fill in the blank in Italian.  Student: what is blue in italian?  Teacher: Can you give me your best guess?  Student: blueo  Teacher: Remember that 'is behind the' is 'e dietro il'  Student: e dietro il blueo cato  Teacher: Hmm... 'is behind the' is 'e dietro il'  Student: e dietro il  Teacher: Hmm... 'cat' is 'gatto'  Student: e dietro il gatto
**Status**: The student initially struggles with the task and asks about the Italian word for "blue," showing a misunderstanding of the translation request. The teacher should guide the student back to the original task by reiterating the need to translate "Is Behind The" into Italian.
**Response:** great but what color is the cat? and who is behind the cat, how do you say bunny? |

Table 20: Prompts and exemplars used in `TPE` for the FoCus dataset

| TPE |
| --- |

**–Thinker–**
**Persona:** You need to analyze the ongoing conversation and carefully infer the internal status exhibited during the conversation about the USER, such as the user's present preferences and status (starting with I know the USER ....), then you need to anticipate the outline of the plan to response the last turn of USER based on the internal status, including the goal of each step and connections between different steps.
**–Planner–**
**Persona:** You should carefully consider a plan in which two external knowledge sources are sequentially called to retrieve evidence for generating the final response step-by-step. Make sure to outline the objectives at each step of the plan and anticipate the content of useful information that may be stored in the corresponding knowledge base. Then, provide a detailed description of the function calls to clarify the process further. For each plan, indicate which external source, along with the source input, is used to retrieve evidence. We can store this evidence in variable So, which can be referenced by subsequent steps. (Plan, So1, Plan, So2, ...)
**Source Set:** The two knowledge sources of evidence are defined as follows:
- PERSONA: This knowledge base stores personal preferences or relevant personal details about the user. It takes in the query and returns a related user persona that assists in addressing the user's current question.
- DOCUMENT: This knowledge base stores background knowledge from Wikipedia as the hint for the given dialogue. Normally, we consider using DOCUMENT when the background knowledge is required and helpful to guide the response generation.
**–Executor–**
**Persona:** Play the role of SYSTEM and generate a helpful response for the following dialogue. To assist you, we provide some corresponding knowledge that might be helpful. Notice that some of this information contains noise so you should trust them with caution.
**Source Knowledge:** retrieved by external retriever.

**Dialogue:** USER: What is the geography of this place? SYSTEM: The Arctic Cordillera is geographically diverse, and much of Ellesmere Island is covered by the Arctic Cordillera, making it the most mountainous in the Canadian Arctic Archipelago. You would love this place since you are interested in geography. USER: What is the overview of this area?
**Thought:** I know that the USER is interested in geography and now he wants to know the overview of The Arctic Cordillera. To respond to this query, I need to search the background knowledge of The Arctic Cordillera and do not require any personal preferences.
**Plan:** Search for more information about the Arctic Cordillera.
**#So1** = DOCUMENT[The Arctic Cordillera]

**Dialogue:** USER: I know this place, but I don't remember the name of this place.
**Thought:** I do not know much about the USER. Since the USER expresses that he knows this place but does not remember. I need to access the knowledge base of personal memory to know the name of this place, and then search for related background knowledge about the place to provide a more informative response.
**Plan:** Search for personal memories about the place.
**#So1** = PERSONA[context]
Plan: Search for more information about the place.
**#So2** = DOCUMENT[So1]

**Dialogue:** USER: Wow, this is amazing! What is this? SYSTEM: This is The Arctic Cordillera located in Canada, which you want to visit. USER: What is the place known for?
**Thought:** I am aware that the USER is interested in visiting The Arctic Cordillera. However, the USER lacks information about the notable features of this location. My task involves researching background knowledge about The Arctic Cordillera and identifying any potential points of interest that align with the user's preferences. Subsequently, I will also need to access a knowledge base containing the USER's personal preferences to enhance the recommendations.
**Plan:** Search for more information about the place.
**#So1** = KNOWLEDGE[The Arctic Cordillera]
**Plan:** Search for personal preferences about the place.
**#So2** = PERSONA[#So1]

Table 21: Prompts and exemplars used in `TPE` for the CIMA dataset

| TPE |
| --- |

**–Thinker–**
**Persona:** You are a teacher who helps a student translate a phrase from English to Italian. You need to infer the student's confusion at the current step and what is the correct direction for the student to the final solution.
**–Planner–** & **–Executor–**
**Persona:** You are a teacher who helps a student translate a phrase from English to Italian. You need to adopt strategies that conform with some educational conversational norms, such as providing hints versus asking questions in appropriate contexts.
**Strategy Set:**
The five strategies are defined as follows:
- Hint: The teacher provides knowledge to the student via a hint.
- Question: The teacher asks a question of the student, which can attempt to determine a student's understanding or continue the conversation.
- Correction: The teacher corrects a mistake or addresses a misconception a student has.
- Confirmation: The teacher confirms a student's answer or understanding is correct.
- Others: Refers to any strategy or approach that does not fall within the predefined categories.

**Dialogue**: Teacher: "Is Inside Of The" is "e dentro la". Please try to fill in the blank in Italian. Student: How do you say blue box in Italian? Teacher: Prepositional phrases separate the two noun phrases. Student: Is it e dentro la box blu?
**Thought:** The student in the current step appears to be somewhat confused about the structure of the translation. While they have made an attempt by saying, "Is it e dentro la box blu?" there is a slight confusion in the word order and article usage. The correct direction for the student is to adjust the word order to "La box blu è dentro" to maintain the correct Italian sentence structure.
**Plan:** Hint
**Do:** box is scatola.
**Plan:** Question
**Do:** Do you remember how to say the plant?

**Dialogue**: Teacher: Green is verde. Please try to fill in the blank in Italian. Student: what is the word for green?
**Thought:** The student in the current step appears to be confused about the Italian word for "green". The correct direction for the student is to use the information provided by the teacher and fill in the blank with "verde," as instructed, to complete the translation.
**Plan:** Hint
**Do:** la pianta e dentro la scatola verdeverde

**Dialogue**: Teacher: 'Is Behind The' is 'e dietro il'. Please try to fill in the blank in Italian. Student: what is blue in italian? Teacher: Can you give me your best guess? Student: blueo Teacher: Remember that 'is behind the' is 'e dietro il' Student: e dietro il blueo cato Teacher: Hmm... 'is behind the' is 'e dietro il' Student: e dietro il Teacher: Hmm... 'cat' is 'gatto' Student: e dietro il gatto
**Thought:** The student initially struggles with the task and asks about the Italian word for "blue," showing a misunderstanding of the translation request. The teacher should guide the student back to the original task by reiterating the need to translate "Is Behind The" into Italian.
**Plan:** Question
**Do:** great but what color is the cat? and who is behind the cat, how do you say bunny?

