# OpenReview forum: "TPE: Towards Better Compositional Reasoning over Conceptual Tools with Multi-persona Collaboration"
_ICLR.cc/2024/Conference — Submitted to ICLR 2024_

### Official Review · Reviewer_89Qo · 2023-10-31

**Soundness:** 2 fair
**Presentation:** 1 poor
**Contribution:** 1 poor
**Rating:** 3
**Confidence:** 4

**Summary:**

The paper presents a prompting method for dialog response generation. The method enables the LLM to incorporate the conceptual tools into its planning and action. It constructs three characters on the same LLM: thinker, planner, and executor. Thinker provides general guidelines; planner decides which conceptual tool to use; and executor composes the final output. Experiments are conducted on two dialog response generation tasks.

**Strengths:**

- The problem of incorporating conceptual tools is interesting and significant.
- The paper compares with various approaches.

**Weaknesses:**

- The paper is poorly written. Many key concepts are not clearly defined. The method description is difficult to follow. Details about the experiment tasks are missing like what these tasks are about and how the evaluation metrics are computed.
- The approach is simply multi-persona prompting, which has been done in various papers (the authors also cited those papers). I do not see a significant novelty compared to previous approaches.
- The approach requires too much human knowledge in designing the prompt. The prompt is highly customized to the problem setting.
- The approach relies entirely on the LLM capability of interpreting the prompt.
- Performance gains compared to baselines are marginal.

**Questions:**

Can you discuss the differences of your method compared to each baseline approach?

Why can't we just use the same approach for functional tools to handle conceptual tools?

---

> ### Author Response · Authors · 2023-11-15
>
> We greatly appreciate your valuable suggestions and feedback. We carefully address your concerns one by one as follows:
>
> Weakness:
>
> 1) **Experimental Details**: We have meticulously defined key concepts, incorporating references from relevant literature. In our experimental design, we deliberately selected multiple source and multiple strategies-grounded response generation tasks, as explicitly outlined in the Introduction and Experiment sections. To enhance clarity, we have included examples and conducted a comprehensive statistical analysis of the corresponding datasets, presented in Figure 1 and the Appendix. In order to provide fair and comparable results, we choose conventional metrics such as BLEU, F1 as shown in Section 4.1 following previous works [1, 2, 3]. We also cite related literature for references.
>
> 2) **Novelty**: Our work firstly introduces conceptual tool, serving as a significant extension and a new perspective to address complex dialogue situations, particularly for multi-source and multi-strategy dialogues. In addition, we also firstly apply  the multi-persona collaboration capability of LLMs into tool learning, while other multi-persona prompting method can not be directly used in this problem. In addition, the collaboration framework is also novel, as a tailored and explainable top-down dialogue planning framework to our targeted problems.
>
> 3) **Expert Knowledge**: We acknowledge that it is necessary to utilize expert knowledge when design the prompts. However, we also need to point out that it is exactly the novelty and values of the proposed method, following lots of other prompting method such as CoT [NeurIPS 2022], ReAct [ICLR 2023], Chameleon. Compared with these baselines, our method achieve better performance and can be applied into other dialogue response generation tasks.
>
> 4) **Performance**: Despite the improvement may not substantial, it is important to consider the complexity of the task, dataset characteristics, and the nature of the baseline models. These two kind of dialogue response generation tasks are challenging and complicated, and we select current SOTA methods including supervised and unsupervised approaches. Illustratively, our method showcases a notable advancement with a remarkable 12 points enhancement in both Avg.B and F1 compared to the supervised state-of-the-art method. Furthermore, in surpassing all other unsupervised approaches grounded on ChatGPT, our method exhibits a superior performance with an approximately 2 points. Thus, we consider our performance improvement is **NOT** marginal. **These results not only highlight the substantial progress achieved by our approach but also underscore the significance of our motivations and innovative solutions in addressing these challenges compared to existing benchmarks. Our contributions extend beyond mere performance metrics, emphasizing the novel perspectives and solutions we bring to the forefront of these problems**. In this way, we believe that the insights gained from our approach are more important for the following reasons: 1. We shows that the LLM can call different conceptual tools automatically and dynamically to solve a complex task; 2) We validate the effectiveness and efficiency (i.e., less cost) of our proposed multi-persona collaboration framework; and 3) The results and analysis suggest that our approach holds the potential to contribute significantly to the advancement and application of dialogue systems within the realms of psychological therapy, educational tutoring and customized dialogue system.
>
>
> More comments are folllowed. Please read them too.

---

> ### Author Response · Authors · 2023-11-15
>
> Questions:
>
> Q1: Can you discuss the differences of your method compared to each baseline approach?
>
> A1: To begin, our proposed TPE (Thinker-Planner-Executor) framework distinguishes itself from all baselines by harnessing the collaborative capabilities of multiple Language Model (LLMs) instances, each embodying distinct personas and playing varied roles in a top-down approach to address complex real-world problems effectively. Furthermore, we focus on elucidating the distinctions between TPE and three representative baselines—Chameleon, ReWOO, and ReAct—to provide a detailed comparative analysis. This approach allows for a clearer understanding of how TPE stands apart from specific baselines, emphasizing its unique contributions and advantages.
>
> 1. Chameleon: As shown in Table 13 and 14, it does not consider the argument dependency between different tools while TPE models the internal relationship such as the PERSONA and DOCUMENT in FoCus dataset. In addition, it can not perform well in multiple strategy dialogue response generation tasks since it generates the response for each strategy independently without considering the relationship, especially for multiple same strategies in single turn.
>
> 2. ReAct: This is a observation-dependent prompting method, which means all previous output must be feed during the output of all intermediate results, resulting in low efficiency and unnecessary cost. It is easily trap into local optima and loop, as shown in ToolQA [4].
>
> 3. ReWOO: To alleviate the problem of ReAct, this method decouples the tool planner and task executor. To better improve the performance and met the specfic requirements of target problems, we make some modifications: 1) Add Thinker modules, which prompt the LLMs generate more additional signals to help the planning. There are two functions: (a) similar with other methods which utilize LLMs to enrich the semantic information of the query in information retrieval [5]; and (b) the intermediate reasoning processing in CoT; 2) Design two variants for different situcations as introduced in Planner and Executor. For example, we observe that as the number of strategies increases, the number of evidence (\#E(number)) can become uncontrollable, making it impractical to present the results of ReWOO on the CIMA and PsyQA datasets. We put the analysis and explanations in the Appendix B.6.
>
> Lastly, it is crucial to highlight that our proposed method consistently outperforms all baselines, striking a balance between efficiency and effectiveness.
>
> Q2: Why can't we just use the same approach for functional tools to handle conceptual tools?
>
> A2. Yes, we can. Actually, we indeed choose these approaches for functional tools as our baselines, such as Chameleon, ReWOO and ReAct. However, as indicated in our main experiments and analysis, our method achieves better performance than these strong baselines under both automatic and human evaluation. There are two reasons as explained in the paper: 1) some of the baselines for functional tools can not perform well when there are multiple calls of the same conceptual tools in the response generation, such as the multiple calls of the same strategy (2nd part of Executor in the Method Section). 2) the introduced top-down multi-persona collaboration framework brings more explainable and controllable generation, resulting in better performance (Experiment and Appendix).
>
> In conclusion, it's crucial to highlight that existing prompting methods do not squarely address our specific target problem—multi-source and multi-strategy grounded response generation tasks. Our contribution extends beyond presenting a better method; fundamentally, we offer a novel perspective that addresses the inherent challenges of our targeted problem. This perspective represents a significant extension of current solutions, emphasizing the unique and valuable contribution our approach brings to the field.
>
> [1] Call for Customized Conversation: Customized Conversation Grounding Persona and Knowledge
>
> [2] Strategize Before Teaching: A Conversational Tutoring System with Pedagogy Self-Distillation
>
> [3] PsyQA: A Chinese Dataset for Generating Long Counseling Text for Mental Health Support
>
> [4] ToolQA: A Dataset for LLM Question Answering with External Tools
>
> [5] Precise Zero-Shot Dense Retrieval without Relevance Labels, ACL 2023

---

> ### Author Response · Authors · 2023-11-20
> **Gentle Reminder**
>
> Dear Reviewer,
>
> We would like to express our gratitude for your valuable time and feedback. Considering that discussion between authors and reviewers is only enabled till 22 Nov, we would appreciate it if you could acknowledge our rebuttal and engage in discussion with us in case you have any further comments.
>
> Best regards,
> All authors of submission 4406

---

> ### Comment · Reviewer_89Qo · 2023-11-21
> **Thanks for your response**
>
> I have trouble understanding 2) of executor. Can you give a concrete example of "multiple identical strategies in a single turn"? an example of "treat these same strategies at different steps independently"? and how does your problem improve upon that?
>
> I am still not perfectly clear about the advantages of your work over ReWoo. How does your planner deal with a large number of strategies? Your plan function takes the tool set as the input. When the set is large, would it be very costly to run the plan function? Why does ReWOO have trouble with a large number of strategies? It seems like in ReWOO, you could easily limit the number of produced plans through prompting.

---

> ### Author Response · Authors · 2023-11-22
>
> Thank you for your timely response and further discussion. Here are some concrete examples and answers to your questions:
>
> 1) **multiple identical strategies in a single turn**: There are many examples in corresponding datasets and we choose one in PsyQA (exactly the introduction example at page 2 in original paper [1]):
>
>      *Question*: The more I think about some things, the more upset I feel. Why?
>
>      *Response*: Hi ~ The more you think about it, the more depressed you feel. This is maybe because you are trapped in ruminant thinking. Ruminant thinking means that ...... Ruminant thinking, as a formof cognition, also has an important effect on emotion. **In this case, you need to calm down first**…For example, you stayed at home and didn't go out for a blind date, and your family said that you just wanted to be single. When you look at it carefully, there is no causal relationship between the two events. ...... But this logic doesn't work. **Of course, in this case, you can also distract your attention to calm yourself down a bit. Take a meditation practice, or go outside to exercise.**
>
>     As shown in the original paper, the highlight parts are generated based on same strategy: `Direct Guidance` in single turn. It is worth noting this case happens more in practice, including but not limited to tutoring dialogue systems and psychological therapy.
>
> 2) **treat these same strategies at different steps independently**: For Chameleon which simply decouples the planning and generation for all tasks without considering the argument dependency, it will generate part of response independently. For example, given the above question, it may generate: strategy\_1, strategy\_2, strategy\_1, strategy\_5 firstly (Here the name and definition of strategies are exactly same as the original dataset as shown in Appendix). Then for each generated plan (strategy here), it will generate the response with question and corresponding strategy as input. Thus, the inter relationship between different strategies is not considered, especially for the multiple same strategies (strategy\_1 in the case).
>
> 3) **how does your problem improve upon that?** We merge the planner and executor to alternatively plan and execute, in order to suit the complex strategy transition in practice, aiming to model the close interaction between the Planner and Executor allows for timely adjustments and responses to intermediate actions. For example, the output should be strategy\_1, part\_1\_response, strategy\_2, part\_2\_response, ...
>
> 4) **Q**: How does your planner deal with a large number of strategies? Your plan function takes the tool set as the input.
>
>    **A**: We regards different strategies as different conceptual tools by feeding the name and definitions at part of prompt (We conduct ablation studies in the paper). It's important to note that our utilization (names and definitions of tools as part of input) aligns with the conventional framework in tool learning. This utilization is consistent with established methods such as Chameleon, ReAct, and ReWOO (as highlighted in Appendix C.8 of the original paper [2]).
>
> 5) **Q**: When the set is large, would it be very costly to run the plan function?
>
>     **A**: Certainly, when dealing with a large set, the computational cost of running the plan function does increase. However, it's important to highlight that our approach is more efficient compared to observation-dependent methods like ReAct. Unlike ReAct, which necessitates the input of the set at each step [2], **our method, akin to other approaches such as Chameleon and ReWOO, only requires the set input at the beginning**. To provide a comprehensive understanding of the computational costs, including token consumption, we have conducted a detailed cost analysis, as outlined in the Appendix, **which validate the better tradeoff between effectiveness and efficient of our method**.
>
> 6) **Q**: Why does ReWOO have trouble with a large number of strategies?
>
>    **A**: It’s important to highlight that as the number of strategies increases, the number of evidence (\#E(number)) in the output of ReWOO can become uncontrollable, making it impractical to present the results. We indicate this at Appendix B.6.
>
>
> More comments are folllowed. Please read them too.

---

> ### Author Response · Authors · 2023-11-22
>
> After all, we want to emphasise the advantages of our proposed method TPE over ReWOO again. First of all, we introduce the multi-persona framework, especially the introduction of Thinker module as detailedly illustrated in last round. We believe that we share a common view, which is the importance and value of multi-persona collaboration as also validated by previous works; Moreover, we enhance the versatility of our approach by designing two variants of TPE, specifically tailored to address more intricate and practical scenarios. These variants expand the applicability of our method, allowing it to effectively handle a broader range of real-world situations.
>
> In the broader context of our contributions, **we assert that our paper represents a pioneering effort in introducing conceptual tools for target problems. By evaluating the performance of current state-of-the-art methods, we believe our work provides a comprehensive and insightful analysis**. Considering the collective discussions and contributions, we humbly contend that our paper stands as the first to systematically introduce and assess the efficacy of conceptual tools for target problems, warranting favorable evaluation.
>
>
> [1] PsyQA: A Chinese Dataset for Generating Long Counseling Text for Mental Health Support, ACL 2021
>
> [2] ReWOO: Decoupling Reasoning from Observations for Efficient Augmented Language Models

---

> > ### Comment · Reviewer_89Qo · 2023-11-23
> > **Thanks for the clarification**
> >
> > I decide to raise my score to 3 but am still leaning towards rejecting the paper. I think the paper definitely has merits but the current form of the paper dilutes those merits. It needs substantial restructuring and that requires another round of review. I suggest the followings:
> >
> > 1. Mathematically formalize the notions of "conceptual tool" and "functional tool" to crisply contrast them. Currently, they are vaguely defined. "Functional tool" is defined through examples (apis, models, programs), and conceptual tool is "a
> > cognitive concept used to help systematic or investigative thought". I do not understand why a list of documents or memories fall into that category. "Conceptual tool" seems just like some text that you can retrieve and add to the LLM prompt.
> >
> > 2. Before introducing your approach, discuss the prior approaches (CoT, ReWOO, etc.) and highlight their disadvantages in your problem (incorporating conceptual tool). This helps the reader understand the motivation and significance of your approach.
> >
> > 3. When introducing your approach, do not just give plain details. Emphasize why addresses the disadvantages of the prior approaches.
> >
> > 4. I had a hard time understanding what the experimental tasks are about, especially the goal of the agent. I'd make the introduction texts more connected with the actual example in Figure 1. Instead of talking in generic terms, I'd try to get the reader to understand what is going on in the example. In the experiment section, I'd define the evaluation metric more clearly (e.g., BLEU, BERT score of what with respect to what?). Additionally, since these tasks do not have a clear definition of success, I think string-matching metrics are insufficient and human evaluation is needed.
> >
> > Thanks for your effort. I hope you find comments helpful to some degree.

---

> ### Author Response · Authors · 2023-11-23
>
> Thank you for your timely actions to raise the score. We totally agree with you regarding 2nd and 3rd comments, and we are in the process of reorganizing the entire paper to adhere to your suggestions and other reviewers. However, we have to clarify more about the 1st and 4th comments since we believe there are still many and deep misundertandings of our paper.
>
> 1: **"Conceptual tool" seems just like some text that you can retrieve and add to the LLM prompt.** We must point it out that conceptual tool is not some text, it is a cognitive concept for humans to understand the world. Specifically, in multi-source dialogue response task (a.k.a, the generation of response requires multiple external sources of knowledge such as persona, document, memory), these different sources serves as conceptual tools. The key concern here is whether or not LLMs can comprehend the distinctions between the concepts of Persona and Document sources (or other sources) and determine the correct order to call them, rather than focusing on the functional tool, such as a retriever (in 3rd paragraph in Introduction section). Besides that, different conversational strategies can also be regarded as conceptual tools, such as *Hint, Question, Correction and so on*. The definition and scope of conceptual tool are totally different from some text that you can retrieve.
>
> 4: **human evaluation**: We calculate the conventional evaluation metrics such BLEU, F1 using generated responses by our proposed methods (or baselines) with ground truth responses in the original datasets. We have reported the results of human evaluation in the Appendix B.1 and indicated it in Section 4.1.
>
> After all, we sincerely appreciate the time and effort you invested in providing valuable suggestions and comments to enhance our work. Your insights are invaluable in helping us refine our paper to better align with the expectations of the conference. Thank you once again for your contributions to our work.

---

### Official Review · Reviewer_X47T · 2023-11-01

**Soundness:** 3 good
**Presentation:** 3 good
**Contribution:** 3 good
**Rating:** 6
**Confidence:** 4

**Summary:**

The paper expand the traditional LLM's tools definition which is mainly for functional tools, to conceptual tools within the context of dialogue systems. Furthermore, the paper introduce a multi-persona collaboration framework: Think-Plan-Execute(TPE). The effectiveness of TPE was demonstrated across multiple dialogue response generation tasks.

**Strengths:**

The paper extend the traditional LLM's definition of functional tools to cognitive tools for dialogue system. Serving a novel perspective to address complex dialogue systems. Based on the cognitive tools, TPE, a tailored and explainable top-down dialogue planning framework is designed to produce more explainable and personalized. Extensive experiments on different datasets are conducted to prove the effectiveness of the proposed framework.

**Weaknesses:**

Some section description of the paper is not very clear to understand,  For example, the formulation part of section 3.2, it's better to be illustrated with some examples.  The cognitive tools seem to lack a formal formulation like the functional tools.

**Questions:**

1. It's a little difficult to understand the formulation symbols in section 3.2, it's better to illustrate with some intuitive example.
2. In this paper's experiment, the cognitive tools seem corresponding the dialogue strategies? Is there more formal definition based on traditional dialogue  theory for it like definition for functional tools? or it changes with different domain dataset?
3. For Table 1's result, Why there is no TPE in the Zero-shot ChatGPT methods? And for CIMA's  F1 score, the ReAct methos is much better than other methods, is there any possible explanation?

---

> ### Author Response · Authors · 2023-11-15
>
> We greatly appreciate your valuable comments and suggestions. Regarding all of the concerns you raised:
>
> Weaknesses:
>
> Thank you for your suggestions, we will definitely add more examples and formulations to better illustrate our method.
>
> Questions:
>
> Q1: It's a little difficult to understand the formulation symbols in section 3.2, it's better to illustrate with some intuitive example.
>
> A1: Please refer to the above answer for weakness.
>
>
> Q2: In this paper's experiment, the cognitive tools seem corresponding the dialogue strategies? Is there more formal definition based on traditional dialogue theory for it like definition for functional tools? or it changes with different domain dataset?
>
> A2. Honestly, the cognitive tools (**conceptual tools** in the paper) corresponds **the multiple knowledge sources, and multiple conversational strategies** in the experiment. We formally define the conceptual tool as a cognitive concept used to help systematic or investigative thought in the Introduction section, and the instances of conceptual tools changes with different domain datasets, as same as functional tools. We indeed borrow some definitions of multi-strategy-grounded dialogue such as psychological therapy and we will follow the suggestions of reviewer U93G to ground the definition with literature in cognitive science.
>
>
> Q3: For Table 1's result, Why there is no TPE in the Zero-shot ChatGPT methods? And for CIMA's F1 score, the ReAct methos is much better than other methods, is there any possible explanation?
>
> A3: The primary challenge associated with zero-shot learning lies in the lack of control over the output, particularly in terms of format. This issue is pervasive across various prompting methods employed in tool learning, including Chameleon, ReWOO, and ReAct. A common drawback shared by these methods is the inability to determine when and how to invoke tools without the incorporation of demonstrations. In this way, all these methods utilize few-shot methods [1, 2, 3]. Ragarding the performance of CIMA, it is worthy noting that there arises a need to employ multiple instances of the same strategy within a single turn response generation, as exemplified in the second part of the Executor, for these multiple strategy dialogues. Notably, in such scenarios, observation-dependent prompting, as exemplified by ReAct, proves to be more advantageous compared to decoupling methods like those seen in Chameleon and ReWOO. This preference stems from the fact that the input for Chameleon and ReWOO remains the same when multiple calls of the same strategies are made. Additional details are provided in the Appendix, and the code implementation can be found in the attached materials.
>
> [1] Chameleon: Plug-and-Play Compositional Reasoning with Large Language Models
>
> [2] ReWOO: Decoupling Reasoning from Observations for Efficient Augmented Language Models
>
> [3] REACT: SYNERGIZING REASONING AND ACTING IN LANGUAGE MODELS

---

### Official Review · Reviewer_U93G · 2023-11-01

**Soundness:** 2 fair
**Presentation:** 1 poor
**Contribution:** 3 good
**Rating:** 5
**Confidence:** 4

**Summary:**

The  paper introduces planning and reasoning framework known as TPE for dialogue systems.  This framework is designed to guide Large Language models in generating more reasonable dialogues. TPE consists of three main components:

 The Thinker - The Thinker attempts to comprehend the current state of the human/persona involved in the conversation. It analyzes the previous conversation to estimate the intent or emotional state of the other persona in the conversation. It then formulates a textual explanation, referred to as a "thought," which describes this state and suggests possible next steps.

The Planner-  The Planner takes the thoughts generated by the Thinker and devises a set of possible actions based on these thoughts, expressed through text.

The Executor -  The Executor executes one of the plans generated by the Planner, resulting in a follow-up dialogue that is presented to the user.

The authors also introduce the concept of "concept tools" in the paper, which is defined as "a cognitive concept used to facilitate systematic or investigative thought."

This paper proposes a framework that decomposes the dialogue generation process into smaller, explainable segments. It compels Large Language Models to first consider the intentions of the person asking the question, similar to a theory of mind approach. This is followed by the generation of a set of strategies that could be useful in fulfilling the user's intentions, and then the creation of conversational text. This approach is distinct from the traditional end-to-end text generation approach.

The authors conducted experiments using two multi-source strategy and dialogue datasets, FoCus and CIMA. They compared their TPE approach applied to Large Language Models (such as Chat GPT and GPT-4) with other approaches, evaluating various metrics including F1 scores, BLEU, BERT score, and ROGUE. The results demonstrate that their approach outperforms other methods across these metrics.

Furthermore, the authors performed an ablation analysis, highlighting the significance of each component within their framework and emphasizing the importance of the Thinker and Planner components in their research.

**Strengths:**

The paper proposes an interesting idea that is to re-enforce thinking and planning through textual information in Large Language Models. Overall, the idea seems reasonable, and important because a method like this will not only enable LLM's to think about intermediate dimensions ( thinking about the user and planning ) before generating a response but also these intermediate dimensions can serve as explanations for the response generated by the user.
The authors also conduct a detailed analysis of why their approach works well on tasks that involve strategic planning through dialogue and a detailed study that show why the the thinker and planner components work

**Weaknesses:**

While the overall picture makes sense, the paper is not adequately presented. To begin with the authors introduce several key components/ definitions that aren't well grounded in the literature. For example, the definition of conceptual tools should be grounded from a more credible source in cognitive science. Further, the paper mentions that the thinker tries to estimate the emotions of the persona involved. Emotions are not defined/referenced well in the paper and can mean a lot of things. A little suggestion. would be to look at the literature on Theory of Mind and try to define the thinker in a manner that aligns with the terms used in the literature .

Further, the I feel the main paper lacks in details about the implementation of thinker, planner, and executor. These details are mentioned in the Appendix. However, I feel that these details should be a part of the main paper.

**Questions:**

1. Are there any technical novelties that the paper proposes?
2. Can there be alternatives in which the thinker can be designed and if so how do they compare with the current approach used in the paper?

---

> ### Author Response · Authors · 2023-11-15
>
> Thank you for taking the time to provide feedback on our paper. We appreciate your constructive comments and suggestions for improvement. We have carefully considered your points and would like to address them in the revised version of the manuscript.
>
>
> Weaknesses:
>
> We appreciate that your acknowledgment about the reasonability and importance of our motivation and the proposed method.
>
> 1) **Cognitive Science Literature**: Upon careful consideration of your comments, we acknowledge the importance of aligning our definitions with established concepts in cognitive science. In our initial draft, we mainly focus on tool learning and refer to definitions from Wikipedia and related literature. However, we understand the necessity of bolstering our literature review with additional citations and aligning our terms with the terms used in related cognitive science. To address this, we are currently in the process of reviewing the papers you suggested for inclusion in the revised manuscript and also others as listed below [1,2,3,4,5,6,7]. Our aim is to incorporate relevant findings from these sources, ensuring a more comprehensive and well-grounded discussion in the cognitive science domain.
>
> 2) **Emotion Definition**: We basically prompt the LLMs to consider the emotions of the users following the definition of previous works [8,9]. We will add the support material in a more clear way.
>
> 3) **Implementation Details**: Thank you for your suggestions. Due to the page limit, we have to put some content in the Appendix, and we will re-organize the paper to meet the page limit and more clear.
>
>
> Questions:
>
> Q1: Are there any technical novelties that the paper proposes?
>
> A1: We posit that our paper stands as a pioneering work, introducing conceptual tools that not only extend the landscape of tool learning but also provide a fresh perspective to tackle intricate dialogue scenarios, including those involving multi-source and multi-strategy dialogues. Besides that, we propose a novel prompting mechanism (top-down multi-persona collaboration framework) to solve these problems following lots of previous prompting methods such as CoT [NeurIPS 2022], ReAct [ICLR 2023], ReWOO and etc, achieving better performance. In light of these contributions, we humbly assert that our motivation and method collectively represent a novel and crucial advancement for the community, particularly in the practical applications of Large Language Models (LLMs).
>
>
> Q2: Can there be alternatives in which the thinker can be designed and if so how do they compare with the current approach used in the paper?
>
> A2: We here simply consider the vanilla prompting method for the Thinker module, to make a fair comparison with current SOTA methods such as ReAct, ReWOO. There are some potential directions which may improve the performance of Thinkers, such as incorporating the knowledge graph or external knowledge sources (e.g., user memory).
>
>
> [1] Does the chimpanzee have a theory of mind?
>
> [2] Theory of mind, Chris Frith and Uta Frith, Cell
>
> [3] The child's theory of mind.
>
> [4] Theory of mind: mechanisms, methods, and new directions
>
> [5] Does the autistic child have a “theory of mind”?
>
> [6] The society of Mind
>
> [7] Theory of Mind Might Have Spontaneously Emerged in Large Language Models
>
> [8] Towards Emotional Support Dialog Systems, ACL 2021
>
> [9] Cue-CoT: Chain-of-thought Prompting for Responding to In-depth Dialogue Questions with LLMs, EMNLP 2023

---

> > ### Comment · Reviewer_U93G · 2023-11-22
> >
> > Thank you for your responses. I have gone through your reviews and reviews by other reviewers and I feel that the paper is not yet up to the mark in terms of the motivation and problem definition. The idea seems interesting. All the best for your future submissions.

---

> ### Author Response · Authors · 2023-11-22
>
> Thank you for your response. We totally understand your concerns regarding the clarity of our motivation and problem definition. However, given the constraints of the page limit, we tried our best to provide a more detailed and comprehensive explanation in the main paper, and put some details in Appendix for references. Currently, we are in the process of reorganizing the entire paper to adhere to your suggestions and those of other reviewers, aiming to present our work in a better manner. We also believe that we share a common objective, which is the continual improvement of our work. We will keep improving the quality of the paper and make corresponding modifications as suggested by ACs and all reviewers, and we are confident that with the necessary improvements, it can be more impactful and aligned with the expectations of the conference. Last but not least, we want to express our gratitude again for your valueable suggestions and comments.

---

### Meta-Review · Area_Chair_tegM · 2023-12-05

**Metareview:**

The paper introduces a framework called, Think-Plan-Execute (TPE), which aims to enhance the planning and reasoning capabilities of Large Language Models (LLMs) in dialogue systems by incorporating conceptual tools. The TPE framework is structured into three components: Thinker, Planner, and Executor, each playing a distinct role in generating dialogue responses. The paper is ambitious in extending the concept of tools in LLMs from functional to conceptual, aiming to improve explainability and controllability in responses. While the idea is innovative and the experimental results show promise, the paper faces criticism for its presentation and clarity. Key concepts are not well-grounded in existing literature, and important details are relegated to appendices. Reviewers also note the lack of novelty compared to existing multi-persona approaches and express concerns about the paper's reliance on highly customized prompts. Overall, the paper presents a significant and interesting idea but falls short in execution and clarity, necessitating further refinement for greater impact in the field.

**Justification For Why Not Higher Score:**

Execution of this paper.

**Justification For Why Not Lower Score:**

NA

---

### Decision · Program_Chairs · 2024-01-16

Reject